# Intensity Profile Projection: A Framework for Continuous-Time Representation Learning for Dynamic Networks

**Alexander Modell**[1]    **Ian Gallagher**[2]    **Emma Ceccherini**[2]    **Nick Whiteley**[2]
**Patrick Rubin-Delanchy**[2]
[1]Imperial College London, U.K.    [2]University of Bristol, U.K.
a.modell@imperial.ac.uk,   {ian.gallagher,emma.ceccherini,
nick.whiteley,patrick.rubin-delanchy}@bristol.ac.uk

## Abstract

We present a new representation learning framework, Intensity Profile Projection, for continuous-time dynamic network data. Given triples $(i, j, t)$, each representing a time-stamped $(t)$ interaction between two entities $(i, j)$, our procedure returns a continuous-time trajectory for each node, representing its behaviour over time. The framework consists of three stages: estimating pairwise intensity functions, e.g. via kernel smoothing; learning a projection which minimises a notion of intensity reconstruction error; and constructing evolving node representations via the learned projection. The trajectories satisfy two properties, known as structural and temporal coherence, which we see as fundamental for reliable inference. Moreoever, we develop estimation theory providing tight control on the error of any estimated trajectory, indicating that the representations could even be used in quite noise-sensitive follow-on analyses. The theory also elucidates the role of smoothing as a bias-variance trade-off, and shows how we can reduce the level of smoothing as the signal-to-noise ratio increases on account of the algorithm 'borrowing strength' across the network.

## 1   Introduction

Making sense of patterns of connections occurring over time is a common theme of modern data analysis and is often approached in one of two ways. On the one hand, we may see dynamic network data as a *graph*, in which connections between the same entities over time are somehow treated as one, e.g. through weighting. This view evokes methodological ideas such as community detection [1, 2], topological data analysis [3], or manifold learning [4, 5]. On the other, we may see the data as a set of *point processes* [6], each modelling the event times of connections between two entities. This view evokes temporal notions such as trend, changepoints and periodicity.

In this paper, we develop a representation learning framework for continuous-time dynamic network data in which ideas from both the graph and temporal domains can be combined. The framework obtains a continuously evolving trajectory for each node which represents its continuously evolving behaviour in the network. These trajectories can be used for data exploration, inference and prediction tasks such as spatio-temporal clustering, behavioural analytics, detecting network trends and periodicities, and forecasting. Our framework provides two basic guarantees which could reasonably be viewed as minimum requirements for these tasks:

1. *structural coherence*: when two nodes exhibit similar behaviour at a point in time, their representations at that time are close;

37th Conference on Neural Information Processing Systems (NeurIPS 2023).

2. *temporal coherence*: when a node exhibits similar behaviours at two points in time, its representations at those times are close.

Despite this, almost all existing embedding algorithms fail to satisfy both properties, and there is a perception that some trade-off between the two is necessary [7, 8]. To our knowledge, unfolded spectral embedding [9, 10] is the only exception, but the method is limited to discrete time.

At its core, our procedure is a spectral method and can be implemented at scale using a sparse singular value decomposition. We give a preliminary motivation for the algorithm as minimising a notion of reconstruction error, before developing more rigorous estimation theory: a non-asymptotic, uniform error bound under minimal assumptions on the data generating process, which draws on recent developments in entrywise eigenvector estimation for random matrices [10–14]. This tight control of the behaviour of estimated trajectories means that even quite noise-sensitive follow-on analyses, such as topological data analysis, could plausibly be shown to be consistent.

With appropriate development, the ideas of this paper could be brought to bear on several problems. The first is building a coherent narrative about the nodes based on an embedding. In contact-tracing data for interactions between children at school (Section 5), we see trajectories mixing during lunch hours and then *returning* to their earlier positions, which represent physical classrooms. Both types of coherence are necessary for this description to be possible. We cannot see this, for example, just by looking at the pattern of clustering over time. Given richer and larger datasets it could be possible to automate this "story-telling" process. Among many possible applications, such a technology could transform how we use contact-tracing data in the future. Second, the jump from discrete to continuous time has substantial implications for mathematical modelling, since it makes notions such as continuity and smoothness possible. This work could pave the way to describing structural dynamics in formal mathematical and quantitative terms, for example, what we mean by "communities splitting" [15], including the *exact time* at which this occurs, or by "network polarisation" [16], and how it could be measured, e.g. as a type of derivative. Finally, in criminal investigations, dynamic networks are often analysed to locate an entity, such as a victim of human-trafficking, whose pseudonym or other identifier is changing [17]. Similarly, in dynamic networks of corporate contracting, it is useful to detect when one company is acting like another in the past, e.g. when a shell company takes over the illegal operations of a sanctioned company [18]. Since two nodes acting the same way, at different epochs, are still embedded to the same position, our framework could lead to novel matching and tracking technologies.

**Related work.** While modelling and performing inference on continuous-time dynamic networks has a well-established literature [6, 19–21], the majority of existing methods for representation learning obtain a single, static representation of each node [22–30], and we only aware of two existing methods which learn continuously evolving node representations from the data we consider [31, 32]. Representation learning for discrete-time dynamic networks has a more established literature, and algorithms broadly fall under community detection methods [33–36], fitting latent position models [37–39], spectral methods [9, 10, 40–42] and word-embedding-based methods [43, 44]. For a specific choice of intensity estimator (the histogram), our method can be viewed as a weighted graph analogue of unfolded spectral embedding [9, 10], a spectral method for discrete-time and multilayer networks, but those papers consider different data and models. Given how limited the options are for handling continuous time, in our method comparison we also include some discrete-time methods which could reasonably be used as alternatives.

## 2 Intensity Profile Projection

**Data.** We consider dynamic network data, denoted $\mathcal{G}$, representing instantaneous undirected interactions between nodes over time, which we define formally as $\mathcal{G} = (\mathcal{V}, \mathcal{E})$ on a time domain $\mathcal{T} = (0, T]$, containing a vertex set $\mathcal{V} = [n]$ and a set of triples $\mathcal{E} = \{(i_e, j_e, t_e)\}_{e \geq 1}$, each corresponding to an undirected interaction event, where $i_e < j_e \in \mathcal{V}, t_e \in \mathcal{T}$. We let $\mathcal{E}_{ij} := \{t : (i, j, t) \in \mathcal{E}\}$ denote the interaction events between nodes $i$ and $j$.

**Model.** We assume the interaction events $\mathcal{E}_{ij}$ are driven by an independent inhomogeneous Poisson process with intensity $\lambda_{ij}(t)$. Informally:

$$\lambda_{ij}(t)\mathrm{d}t = \mathbb{P}\left\{\text{interaction between nodes } i \text{ and } j \text{ in } (t, t + \mathrm{d}t]\right\}.$$

---

**Algorithm 1** Approximate Intensity Profile Projection

---

**Input:** Continuous time dynamic graph $\mathcal{G}$, dimension $d$.

1: Construct intensity estimates $\widehat{\lambda}_{ij}(\cdot)$ of $\lambda_{ij}(\cdot)$ from $\mathcal{E}_{ij}$ for all $i < j$.
2: Compute the top $d$ left singular vectors $\widehat{u}_1, \ldots, \widehat{u}_d$ of

$$\begin{bmatrix} \widehat{\boldsymbol{\Lambda}}(t_1) & \widehat{\boldsymbol{\Lambda}}(t_2) & \cdots & \widehat{\boldsymbol{\Lambda}}(t_B) \end{bmatrix}$$

where $t_1 < \cdots < t_B$ are equally spaced points on $(0, T]$.
3: Define the trajectory of node $i$ as

$$\widehat{X}_i(t) := \widehat{\mathbf{U}}_d^\top \widehat{\Lambda}_i(t)$$

where $\widehat{\mathbf{U}}_d := (\widehat{u}_1, \ldots, \widehat{u}_d)$.

**Output:** Node trajectories $\widehat{X}_1(t), \ldots, \widehat{X}_n(t)$.

---

We represent these intensities in a symmetric time-varying matrix $\boldsymbol{\Lambda}(\cdot) : \mathcal{T} \to \mathbb{R}_+^{n \times n}$.

**Procedure.** Intensity Profile Projection can be summarised as follows.

1. **Intensity estimation.** Construct intensity estimates $\widehat{\lambda}_{ij}(\cdot)$ of $\lambda_{ij}(\cdot)$ from $\mathcal{E}_{ij}$ for all $i < j$.

2. **Subspace learning.** Compute the top $d$ eigenvectors $\widehat{\mathbf{U}}_d = (\widehat{u}_1, \ldots, \widehat{u}_d)$ of

$$\widehat{\boldsymbol{\Sigma}} := \frac{1}{T} \int_0^T \widehat{\boldsymbol{\Lambda}}^2(t) \mathrm{d}t, \tag{1}$$

where $\widehat{\boldsymbol{\Lambda}}(t)$ has symmetric entries $\widehat{\lambda}_{ij}(t)$, and rows denoted $\widehat{\Lambda}_i(t)$ called *intensity profiles*.

3. **Projection.** For a query node $i$ at time $t$, project the intensity profile $\widehat{\Lambda}_i(t)$ onto the subspace spanned by $\widehat{u}_1, \ldots, \widehat{u}_d$, to obtain $\widehat{X}_i(t) = \widehat{\mathbf{U}}_d^\top \widehat{\Lambda}_i(t)$.

While we develop more principled statistical justifications for the procedure in future sections, it is inspired by a simple reconstruction argument. For an arbitrary $d$-dimensional subspace spanned by the orthonormal columns of a matrix $\mathbf{V}_d \in \mathbb{R}^{n \times d}$, let

$$\widehat{r}_i(t; \mathbf{V}_d) := \left\| \mathbf{V}_d \mathbf{V}_d^\top \widehat{\Lambda}_i(t) - \widehat{\Lambda}_i(t) \right\|_2$$

denote the reconstruction error of node $i$ at time $t$, and define the *integrated residual sum of squares* as

$$\widehat{R}^2(\mathbf{V}_d) := \int_0^T \sum_{i=1}^n \widehat{r}_i^2(t; \mathbf{V}_d) \, \mathrm{d}t.$$

**Lemma 1.** *Among all $d$-dimensional subspaces of $\mathbb{R}^n$, the column span of $\widehat{\mathbf{U}}_d$ minimises the integrated residual sum of squares criterion $\widehat{R}^2$.*

Lemma 1 may be viewed as a dynamic analogue to the classical Eckart-Young theorem on low-rank matrix approximation [45]. A proof is given in Section E of the appendix.

### 2.1 Intensity estimation

The choice of intensity estimator is left fully open, but our theory makes two important recommendations. First, there are computational gains to be made using sparse estimators for subspace learning. Second, the procedure borrows strength across the network, and can give precise representations even when the individual intensity estimates are noisy (e.g. inconsistent). In our experiments, we focus on standard non-parametric estimators such as the histogram or kernel smoothers, and choose kernels with finite support to induce sparse estimates.

### 2.2 Subspace learning

The subspace learning step of our procedure involves the computation of an integral, and computing the eigendecomposition of the resulting dense matrix $\widehat{\boldsymbol{\Sigma}}$, both of which may be infeasible for large

networks. If a sparse intensity estimator is employed in step 1 of the procedure and we approximate the integral (1) using a numerical quadrature scheme, then step 2 can be rephrased as a single sparse, truncated singular value decomposition, which can be computed quickly for very large networks using a efficient solver [46, 47].

Consider the numerical approximation

$$\widehat{\boldsymbol{\Sigma}} \approx \frac{1}{B} \sum_{b=1}^{B} \widehat{\boldsymbol{\Lambda}}^2(t_b) \tag{2}$$

where $t_1 < \cdots < t_B$ are equally spaced points on $(0, T]$. The top $d$ eigenvectors of the right-hand-side of (2) are then equal[1] to the top $d$ left singular vectors of the matrix

$$\left[\widehat{\boldsymbol{\Lambda}}(t_1) \ \widehat{\boldsymbol{\Lambda}}(t_2) \ \cdots \ \widehat{\boldsymbol{\Lambda}}(t_B)\right], \tag{3}$$

the row concatenation of $\widehat{\boldsymbol{\Lambda}}(t_1), \ldots, \widehat{\boldsymbol{\Lambda}}(t_B)$. This procedure is presented in Algorithm 1, and we discuss its computational complexity in Section D of the appendix.

## 2.3 Projection

The inductive nature of the Intensity Profile Projection allows us to obtain representations $\widehat{X}_i(t)$ on demand, for example, the full trajectory for a particular node, or the representations of the entire graph at a point in time. It is possible to obtain representations for intensity profiles outside the training sample, corresponding to new nodes or times outside the training domain, allowing online inference. In practice, one will need to retrain occasionally, i.e. return to step 2, although we leave the discussion of this computational and statistical trade-off for future work (see, for example, [48] in the context of static networks).

## 3 Estimation theory

In this section, we develop estimation theory showing the sense in which $\widehat{X}_i(t)$ is a "good" estimator of $X_i(t) := \mathbf{U}_d^\top \Lambda_i(t)$ where $\mathbf{U}_d = (u_1, \ldots, u_d) \in \mathbb{R}^{n \times d}$ is the matrix containing the top-$d$ orthonormal eigenvectors of

$$\boldsymbol{\Sigma} := \frac{1}{T} \int_0^T \boldsymbol{\Lambda}^2(t) \mathrm{d}t.$$

In this section, we assume, without loss of generality, that $\mathcal{T} = (0, 1]$. We now introduce some quantities which appear in our main theorem. Firstly, we assume that each $\lambda_{ij}(\cdot)$ is Lipschitz with constant $L$, and is upper bounded by $\lambda_{\max}$. Secondly, we define the (reduced) condition number and the eigengap,

$$\kappa := \frac{\sigma_1}{\sigma_d}, \qquad \text{and} \qquad \delta := \sigma_d - \sigma_{d+1},$$

respectively, where $\sigma_1^2 \geq \cdots \geq \sigma_n^2$ are eigenvalues of $\boldsymbol{\Sigma}$. Finally, we introduce the subspace coherence parameter

$$\mu := \sqrt{\frac{n}{d}} \, \|\mathbf{U}_d\|_{2,\infty},$$

which is small when, informally, information about a single entry of $\boldsymbol{\Sigma}$ is "spread out" across the matrix [49].

Rather than attempt to develop a theoretical framework encompassing all intensity estimators, we choose arguably the most rudimentary, the histogram, and we expect more powerful estimators will only improve matters. This choice of estimator is also attractive because it allows us pinpoint the crucial practical considerations at play.

---

[1]Up to signs, rotations in the eigenspaces in the case of repeated eigenvalues, and assuming a gap between the $d$th and $(d + 1)$th eigenvalues.

**Notation.** We say an event $E$ occurs *with overwhelming probability* if $\mathbb{P}(E) \geq 1 - n^{-c}$ for any constant $c > 0$. We use $a \lesssim b$ to denote that $a \leq Cb$ where $C > 0$ is a universal constant which, when qualified with the prior probabilistic statement, may depend on the constant $c$. Additionally, we write $a \asymp b$ if $a \lesssim b$ and $a \gtrsim b$.

We now state the assumptions we require for our theorem. Our first assumption is that the intensities are bounded.

**Assumption 1** (Bounded intensities). The intensities are upper bounded by a constant which doesn't depend on the other quantities in the problem; i.e. $\lambda_{\max} \lesssim 1$.

Our second assumption is on the population integrated residuals. It ensures that the intensity profiles $\Lambda_1(t), \ldots, \Lambda_n(t)$ do not deviate "too much" from a common low-dimensional subspace.

**Assumption 2** (Small population residuals). The population residuals satisfy

$$r_1(t), \ldots, r_n(t) \lesssim \sqrt{\frac{d}{n}} \mu \delta \log^{5/2} n$$

for all $t \in [0, 1]$.

Our third assumption is a technical condition on the eigengap which, broadly speaking, ensures that there is "enough signal".

**Assumption 3** (Enough signal). The eigengap satisfies $\delta \log(\delta/\sqrt{n\lambda_{\max}}) \gtrsim \kappa n \lambda_{\max}$.

Our final assumption is on the bin size, and ensures that the bins are not chosen "too small".

**Assumption 4** (Large enough bins). The number of bins satisfies $M \lesssim n\lambda_{\max}/\log^3 n$.

These assumptions are weaker than those typically required in the literature (e.g. on stochastic block models and random dot product graphs [2, 50]). To emphasise this point, consider the following stronger alternative assumptions which imply Assumptions 1, 2 and 3:

**Assumption 1a.** The intensities $\lambda_{ij}(t)$ are of comparable order, i.e. $\lambda_{ij}(t) \asymp \rho$ for some $\rho \lesssim 1$ and all $i, j \in [n], t \in (0, 1]$.

**Assumption 2a.** The matrix $\mathbf{\Sigma}$ has rank $d \asymp 1$; is incoherent, i.e. $\mu \asymp 1$; and its non-zero eigenvectors are of comparable order, i.e. $\sigma_1 \asymp \sigma_d > \sigma_{d+1} = 0$.

It is immediate that Assumption 1a implies Assumption 1 and under Assumption 2a, the population residuals are all exactly zero, $\kappa \asymp 1$ and $\delta \asymp n\rho$, which implies Assumptions 2 and 3.

Assumption 4 requires that the expected number of events involving each node in each bin is at least of the order $\log^3 n$. This is analogous to the $\log n$ degree growth required for perfect clustering under the binary stochastic block model. Since the latter is an information-theoretic bound [51] and the additional logarithmic powers in our work stem from the sub-exponential tails of the Poisson distribution, we do not think this assumption can be weakened.

We now state our main theorem, which under Assumptions 1-4, provides a non-asymptotic bound on the error between the learned representations and their population counterparts, which holds uniformly over the whole node-set and the time domain.

**Theorem 1.** *Suppose that* $\widehat{\lambda}_{ij}(t)$ *are histogram estimates with $M$ equally-spaced bins and that Assumptions 1-4 hold. Then with overwhelming probability, there exists an orthogonal matrix* $\mathbf{W}$ *such that*

$$\max_{i \in [n]} \sup_{t \in (0,1]} \left\| \mathbf{W}\widehat{X}_i(t) - X_i(t) \right\|_2 \lesssim \frac{n^{3/2}L\lambda_{\max}}{M\delta} + \mu\sqrt{M\lambda_{\max}d} \cdot \log^{5/2} n. \tag{4}$$

The proof of Theorem 1 is given in Section F of the appendix. As a corollary to Theorem 1, we state a simplified version of this result in which we replace Assumptions 1, 2 and 3 with the stronger Assumptions 1a and 2a. Since the Lipschitz constant $L$ scales with the order of the intensities, and we define the quantity $L_0$ satisfying $L = \rho L_0$ which is invariant to the rescaling of intensities.

**Corollary 1.** *Suppose that* $\widehat{\lambda}_{ij}(t)$ *are histogram estimates with $M$ equally-spaced bins and that Assumptions 1a, 2a and 4 hold. Then with overwhelming probability, there exists an orthogonal matrix* $\mathbf{W}$ *such that*

$$\max_{i \in [n]} \sup_{t \in (0,1]} \left\| \mathbf{W}\widehat{X}_i(t) - X_i(t) \right\|_2 \lesssim \frac{n^{1/2}\rho L_0}{M} + \sqrt{M\rho} \cdot \log^{5/2} n. \tag{5}$$

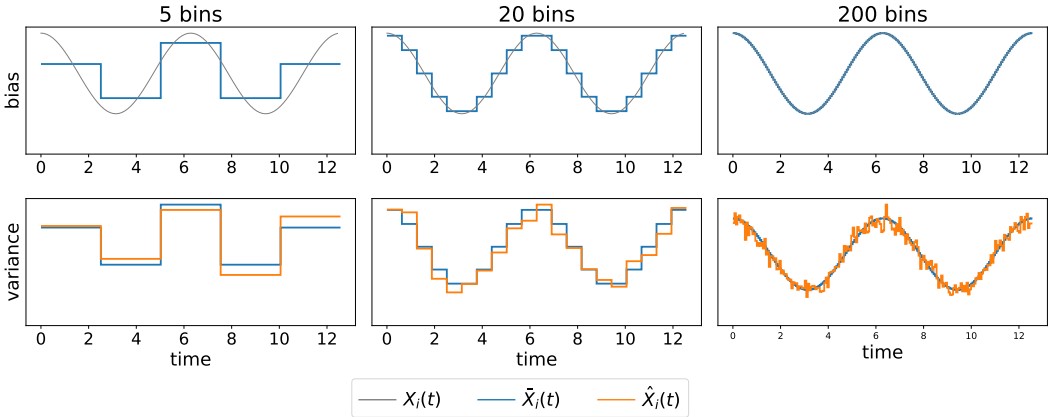

Figure 1: A bias-variance trade-off. We simulate a network with common intensities $\lambda_{ij}(t) = 0.7 \times \{2 + \cos(t)\}$ for all $i, j$, and apply Intensity Profile Projection with a histogram intensity estimator with 5, 20, and 200 bins. In the 'bias' plots, the gray lines shows an estimand $X_i(t)$, while the blue lines shows its histogram approximation. The discrepancy between the gray line and the blue line corresponds the bias of the Intensity Profile Projection estimator. In the 'variance' plots, the blues lines are as in the 'bias' plots and the orange line show the estimate obtains using Intensity Profile Projection into one dimension. The discrepancy between the blue line and the orange line corresponds the variance of the Intensity Profile Projection estimator.

### 3.1 A bias-variance trade-off

The first term in the bound corresponds to the bias between $\bar{X}_i(t)$ and $X_i(t)$, where $\bar{X}_i(t)$ is a histogram approximation to $X_i(t)$ (modulo orthogonal transformation, see Section F of the appendix). The second term corresponds to the variance of the estimate.

Theorem 1 gives some theoretical guidance on how to select the number of bins in the histogram estimator. For simplicity, we consider the setting of Corollary 1. Ignoring logarithmic terms in $n$, the bound in (5) is optimised by choosing

$$M \asymp \left(n\rho L_0^2\right)^{1/3}.$$

Figure 1 illustrates this bias-variance trade-off with an example. We simulate a dynamic network with 100 nodes with common intensities $\lambda_{ij}(t) = 0.7 \times \{2 + \cos(t)\}$, for all $i, j$, on the time domain $(0, 4\pi]$.

The top row shows the population representation $X_i(t)$ of a single node (gray) and its histogram approximation $\bar{X}_i(t)$ (blue) for a variety of bin sizes. The more bins that are chosen, the smaller the bias and the more $\bar{X}_i(t)$ resembles $X_i(t)$. The bottom rows shows the histogram approximation $\bar{X}_i(t)$, and the estimate $\widehat{X}_i(t)$ (orange) obtained using Intensity Profile Projection. The fewer bins that are chosen, the smaller the variance and the more that $\widehat{X}_i(t)$ resembles $\bar{X}_i(t)$.

## 4 Structural and temporal coherence

For many practical inference tasks, it is desirable for a representation learning procedure to possess the following two properties:

- **Structural coherence.** If two nodes exhibit statistically indistinguishable behaviour at a given time, then their representations at that time are close. That is, if $\Lambda_i(t) = \Lambda_j(t)$, then $\widehat{X}_i(t) \approx \widehat{X}_j(t)$;
- **Temporal coherence.** If a node exhibits statistically indistinguishable behaviour at two distinct points in time, then its representations at both these times are close. That is, if $\Lambda_i(s) = \Lambda_i(t)$, then $\widehat{X}_i(s) \approx \widehat{X}_i(t)$.

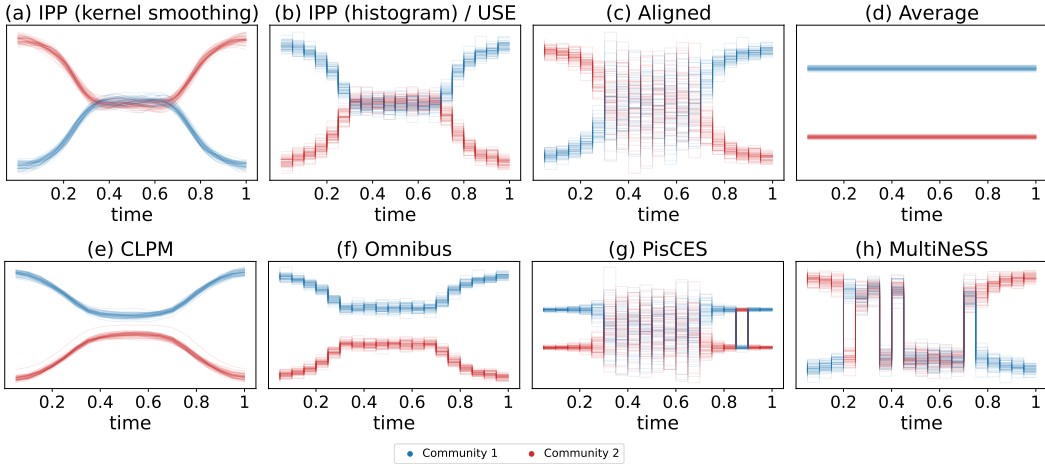

Figure 2: One-dimensional PCA visualisation of the two-dimensional node representations, obtained using a collection of methods, for a network simulated from a bifurcating block model. Colours correspond to the community membership of the node.

It has been observed in a recent survey of [7] that almost all existing dynamic network embedding procedures possess only one of these properties, but not both.

In the following lemma, we formally define $\widehat{X}_i(s) \approx \widehat{X}_j(t)$ to mean that $X_i(s) = X_j(t)$, referring to equality "up to statistical noise" in the sense of Theorem 1.

**Lemma 2.** *Intensity Profile Projection is both structurally and temporally coherent.*

This follows from the simple observation that $X_i(t)$ is a fixed function of $\Lambda_i(t)$ for all $i$ and $t$. To the best of our knowledge, Intensity Profile Projection is the only existing continuous-time procedure which satisfies these desiderata.

### 4.1 Simulated example: a bifurcating block model

To illustrate these properties, we simulate a two-community dynamic stochastic block model (i.e. where $\Lambda(t)$ is block structured) in which the intra-community intensities and inter-community intensities are initially distinct, they then gradually merge, remain indistinguishable for some time, and finally diverge. We refer to this model as a *bifurcating block model* and provide full details of the simulation in Section A of the appendix.

We apply Intensity Profile Projection to the simulated network, using both a histogram intensity estimator, and a kernel smoother, to produce two-dimensional representations. For visualisation, we reduce the dimension from two to one using a dynamic adaptation of principal component analysis (see Section C of the appendix), and the resulting representations are shown in Figures 2(a) and (b).

In both cases, the estimated trajectories mirror the underlying dynamics of the network: the two communities are in well separated to begin with, gradually merge, remain relatively constant before returning to the positions in which they started.

We now illustrate the potential pitfalls of some more naive approaches for embedding dynamic networks. We find that most existing methodology can be viewed as some combination of the two techniques:

- **Alignment** [42]. Obtain a sequence of static snapshots of the network, embed each of the networks snapshots separately and subsequently align the embedding from window $t + 1$ with the embedding from window $t$.

- **Averaging** [25]. Obtain a static summary of the network by averaging it over time, and to embed this to obtain constant node representations.

Alignment is structurally coherent, however can fail to be temporally coherent. Averaging is temporally coherent, but can fail to be structurally coherent. To illustrate this point, we apply both approaches, using adjacency spectral embedding into two dimensions, orthogonal Procrustes alignment and linear interpolation, to a network simulated from the bifurcating block model. Figures 2(c) and (d) show visualisations of the trajectories obtained from each approach.

## 4.2 Method comparison

In this section, we demonstrate how our procedure compares to some existing methods on the simulated data described above. Due to the limited number of continuous-time methods, we include a number of discrete-time methods (Omnibus, PisCSE and MultiNeSS) which we give as an input a discrete sequence of snapshots $\mathbf{A}(1), \ldots, \mathbf{A}(M)$ of our simulated continuous-time networks. We compare the following methods:

- **IPP (kernel smoothing)**. Algorithm 1 applied with intensities estimated using kernel smoothing.
- **IPP (histogram) / USE** [10]. Algorithm 1 applied with intensities estimated using a histogram estimator. Equivalent to a weighted extension of the Unfolded Spectral Embedding algorithm of [10].
- **CLPM** [31]. Fits a continuous latent position model $\log \lambda_{ij}(t) = \beta - \|Z_i(t) - Z_j(t)\|^2$ with a penalty on large velocities in the latent space.
- **Omnibus** [40]. Approximately factorises the matrix $\mathbf{A}$ with blocks $\mathbf{A}[k, l] = \frac{1}{2}(\mathbf{A}(k) + \mathbf{A}(l))$, using a spectral decomposition.
- **PisCES** [41]. Minimises the objective function

$$\sum_{k=1}^{M} \|\mathbf{L}(k) - \mathbf{L}^\star(k)\|_F^2 + \alpha \sum_{k=1}^{M-1} \|\mathbf{L}^\star(k) - \mathbf{L}^\star(k+1)\|_F^2,$$

for $\mathbf{L}^\star(1), ..., \mathbf{L}^\star(M)$, where $\alpha \in [0, 1]$ and $\mathbf{L}(k)$ are the Laplacian normalisations of $\mathbf{A}(k)$. Then, approximately factorises each $\mathbf{L}^\star(1), ..., \mathbf{L}^\star(M)$ using spectral decompositions.
- **MultiNeSS** [52]. Fits a latent position model $\mathbf{A}_{ij}(k) \sim Q\{\cdot; f(Z_i(k), Z_j(k)), \phi\}$, where $Q(\cdot; \theta, \phi)$ is a parametric distribution.

We use an embedding dimension of $d = 2$ for all methods, and for visualisation we reduce this to one using PCA. Additional details such as hyperparameter selection, where applicable, are given in the Section A of the appendix.

The CLPM and Omnibus methods produce representations which are temporally coherent, however both fail to capture the complete merging of the communities, shown by Figures 2(e) and (f), and are therefore not structurally coherent. The PisCES and MultiNeSS methods produce representations which are structurally coherent, however both are unstable when the communities are indistinguishable, shown by Figures 2(g) and (h), and are therefore not temporally coherent.

## 5 Real data

We demonstrate Intensity Profile Projection on a dataset containing the face-to-face interactions of the pupils of a primary school in Lyon over two days in October 2009 [53]. During the study, discreet radio-frequency identification devices were worn by 232 pupils and 10 teachers which recorded their face-to-face interactions. When two participants were in close proximity over an interval of 20 seconds, the timestamped interaction event was recorded. The school contains five year groups, each divided into two classes, and each class has an assigned room and an assigned teacher. The school day runs from 8:30am to 4:30pm, with a lunch break from 12:00pm to 2:00pm, and no data was gathered on contacts taking place outside the school or during sports activities. For more details about the study and dataset, we refer the reader to [53].

We apply Intensity Profile Projection to the data corresponding to each day of the study using a kernel smoother with an Epanechnikov kernel, choosing a bandwidth of 5 minutes and computing 30 dimensional trajectories.

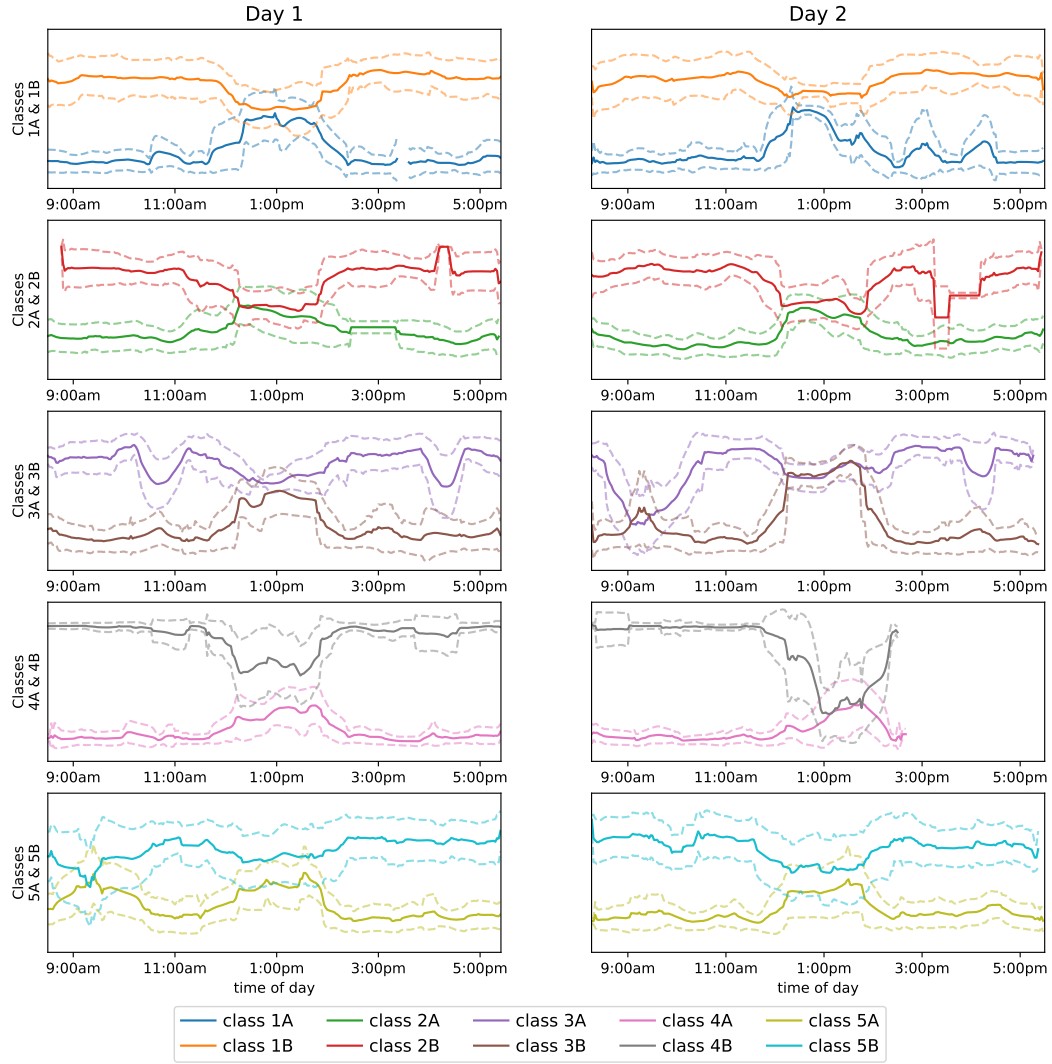

Figure 3: One-dimensional PCA visualisation of the 30-dimensional node representations for pairs of classes in the same year group. The solid lines show the average trajectory for each class, and the dashed line show one standard deviation above and below.

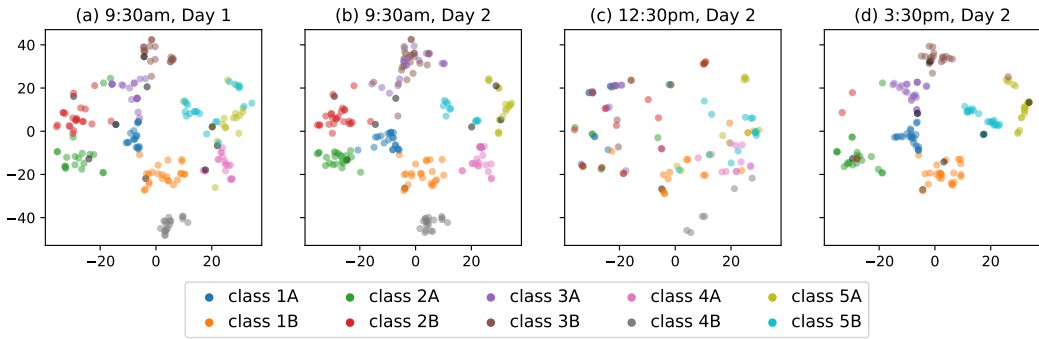

Figure 4: Two-dimensional t-SNE visualisation of the 30-dimensional node representations of all pupils and teachers evaluated at 9:30am on Day 1, and 9:30am, 12:30pm and 3:30pm on Day 2.

To visualise the node trajectories, we first rescale them to have unit norm, which has the effect of removing information about the "activeness" of a node from its representation (see, for example, [54]), and apply two dimension reduction techniques. The first is principal component analysis (PCA), which we adapt to our dynamic setting by projecting the (centered) representations onto the the direction of maximum average variance over the time domain. This visualisation gives us a temporally coherent view on the trajectories (more details are given in Section C of the appendix). In Figure 3, we visualise the trajectories of each pair of classes in each year group using PCA, and for clarity, we just plot the average trajectory for each class, along with one standard deviation above and below.

The second is t-Distributed Stochastic Neighbor Embedding (t-SNE), a popular non-linear dimension-reduction tool which provides enough flexibility to visualise the whole set of representations at each point in time. Figure 4 shows t-SNE visualisations of the node representations at a collection of times throughout the study. In Section B of the appendix, we include analogous figures for aligned spectral embedding and Omnibus embedding for comparison.

Figure 3 clearly shows the mixing of classes during the lunch hours, and from Figures 4, we see that the the representations are much more fragmented during the lunch hour (12:30pm, Day 2) than they are during lessons at the other times, where they form tighter clusters corresponding to classes.

While it is reassuring that the geometry of the trajectories reflects the *known* class and timetable structures of the school, it also allows us to uncover structure in the data that *was not known* from the report on the study. For example, classes 5A and 5B (olive and cyan, respectively) merge into a single cluster at approximately 9:30am on Day 1, and classes 3A and 3B (brown and pink, respectively) do the same at approximately 9:30am on Day 2. One might conjecture that this corresponds to a joint lesson, which is taken by the students of both classes in a year group.

# 6 Discussion

We have presented an algorithmic framework to learn continuous-time, low-dimensional trajectories representing the evolving behaviours of nodes in a dynamic network. We view our framework as providing a platform on which novel inference procedures can be developed, particularly combining graph and temporal concepts. For example, in dynamic networks with continuously evolving community structure, it might be interesting to develop procedures for detecting branching points (see bifurcating block model example, Section 4), or measures of polarisation and cohesion in the network via the velocities of the trajectories. More generally, we believe there is much left to understand and exploit in the time-evolving topology and geometry of these representations.

A limitation of our framework is the need for bandwidth and dimension selection. These decisions are difficult because they are trade-offs, bias versus variance in the case of bandwidth selection (as seen here), and statistical versus computational in the case of dimension selection (see e.g. [55]). In the presence of a specific supervised downstream task, both decisions could be assisted by cross-validation. In unsupervised settings with reasonably-sized networks, our method is very fast, allowing expedient exploration of different choices.

Our theory suggests selecting a dimension which corresponds to an "eigengap" in the spectrum. In practice, it is possible than the spectrum does not decay quickly and no eigengap is present. This likely corresponds to the violation of Assumption 3. Some possible solutions are to take an entrywise transformation of the intensity profiles, such as the square root, to temper heavy tails [56], or to employ a robust subspace estimator, such as robust PCA [57].

Our method might be viewed as a dynamic analogue of adjacency spectral embedding for static graphs [58] and, as a result, in future research it could be profitable to find dynamic analogues of other variants of spectral embedding, e.g. applying Laplacian normalisation [59–61] or regularisation [54, 62].

We believe improved dynamic network analysis can be used for societal good, in applications such as cyber-security, or combating human-trafficking, fraud, and corruption. However, one should also be aware of the risks, particularly to individual privacy and targeted influence.

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

# Appendix to "Intensity Profile Projection: A Framework for Continuous-Time Representation Learning for Dynamic Networks"

## A    Details of the simulated example and method comparison of Sections 4.1 and 4.2

We simulate data according to the following generative process, which might be viewed as describing as a dynamic, two-community stochastic block model. Assign to each node $i$ a variable $z_i \in \{1, 2\}$ denoting its community (which does not change with time). If nodes $i$ and $j$ are in the same community, i.e., $z_i = z_j$, the point process $\mathcal{E}_{ij}$ follows a homogeneous Poisson process with (fixed) intensity $\eta_0$. Otherwise, $\mathcal{E}_{ij}$ follows an inhomogeneous Poisson process with intensity

$$\lambda_{ij}(t) = \begin{cases} \eta_0 \exp\{\eta_1(t - s_1)\} & t < s_1, \\ \eta_0 & s_1 \le t < s_2, \\ \eta_0 \exp\{-\eta_1(t - s_2)\} & t \ge s_2, \end{cases}$$

where $0 < s_1 < s_2 < T$. This model describes two communities gradually coming together until fully merging by time $s_1$, before splitting at time $s_2$ and then gradually drifting apart. We simulate from this model using the parameters $T = 1$, $n = 100$, $z_1, \ldots, z_{50} = 1$, $z_{51}, \ldots, z_{100} = 2$, $\eta_0 = 100$, $\eta_1 = 10$, $s_1 = 0.3$ and $s_2 = 0.7$.

In our method comparison, we used an embedding dimension of $d = 2$ for all methods, unless otherwise stated. For the discrete-time methods, we construct a series of 20 snapshots of the continuous-time network, each a weighted static network whose edge weights are the number of events which occur on the edge in the corresponding time window. The selection of hyperparameters for each method is outlined below:

- **Intensity Profile Projection (histogram)**: We used a bin size of $\frac{1}{M} = \frac{1}{20}$.

- **Intensity Profile Projection (kernel smoothing)**: We used a Epanechnikov kernel with bandwidth 0.1 and applied the approximate Intensity Profile Projection algorithm with $B = 20$. Different values of bandwidth gave similar results in terms of embedding structure; we chose this bandwidth to achieve the desired smoothness.

- **CLPM** [31]: The dimension is automatically computed by the algorithm as $d = 2$. The hyperparameters are chosen equal to the ones used in "Simulation C" in [31] which is a similar simulated example with two communities. We used 19 changes point which correspond to 20 windows. The implementation was obtained from the Github repository `https://github.com/marcogenni/CLPM`.

- **PisCES** [41]: The dimension is automatically selected by the algorithm as $d = 2$. The smoothing parameter is chosen with cross-validation which results in equivalent log-likelihood values for $\alpha$ from 0.00001 to 0.001. We choose $\alpha = 0.001$ which is the larger value for which the algorithm converges. The implementation was obtained from the Github repository `https://github.com/xuranw/PisCES`.

- **MultiNeSS** [52]: The dimension is automatically selected by the algorithm as $d = 2$ for all windows except windows 5 to 16 for which $d = 1$ is selected. For these windows, we set missing the second dimension to zeros. The implementation is obtained from the `multiness` R package (available on CRAN) and hyperparameters are set to their default values.

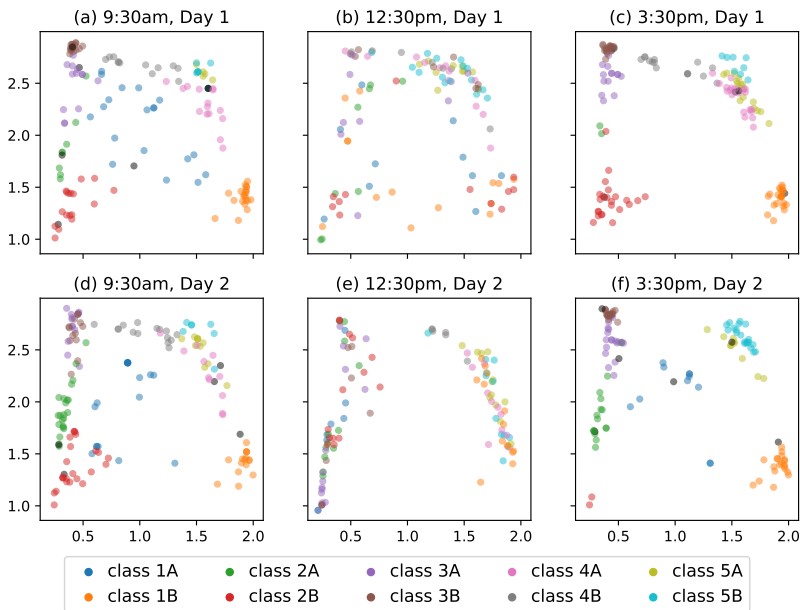

Figure 5: The first two dimensions of the spherical coordinates of the coordinates $\widehat{X}_i(t)$ using the histogram intensity estimator for times corresponding to the morning, lunchtime and afternoon across both days. The colours indicate classes with black points representing teachers.

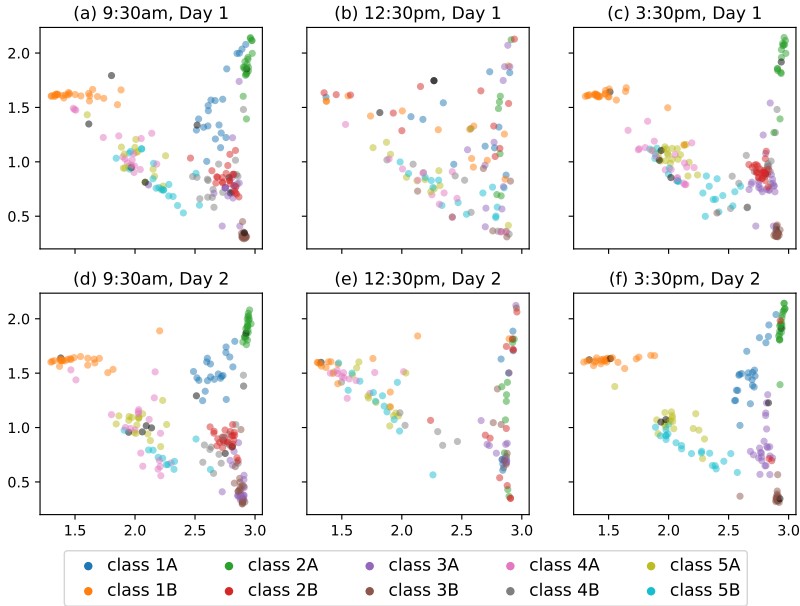

Figure 6: The first two dimensions of the spherical coordinates of the trajectories $\widehat{X}_i(t)$ using the Epanechnikov kernel smoother for times corresponding to the morning, lunchtime and afternoon across both days. The colours indicate classes with black points representing teachers.

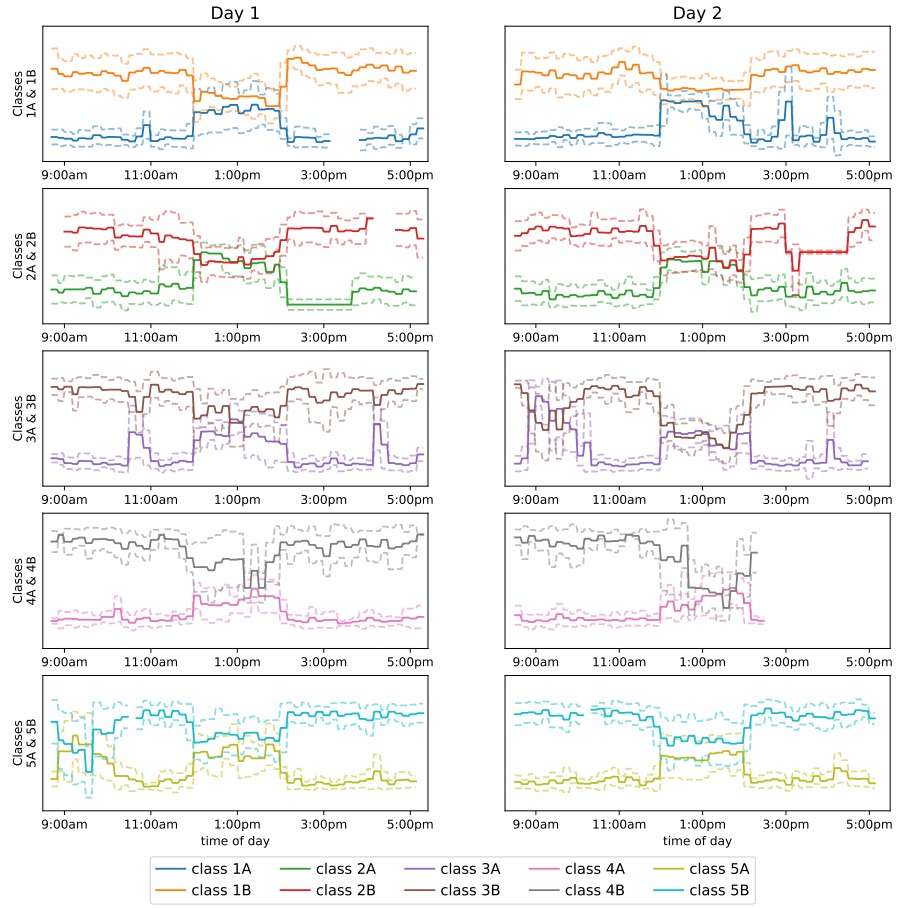

Figure 7: One-dimensional PCA visualisation of the 30-dimensional node representations for pairs of classes in the same year group. The solid lines show the average trajectory for each class, and the dashed line shows one standard deviation above and below.

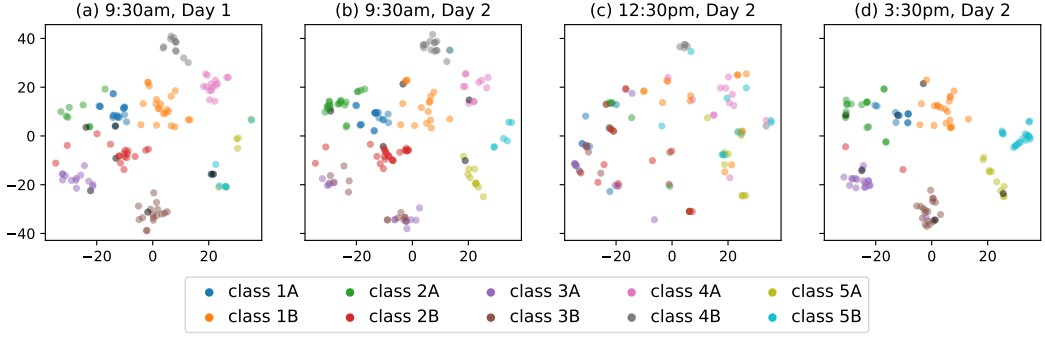

Figure 8: Two-dimensional t-SNE visualisation of the 30-dimensional node representations of all pupils and teachers evaluated at 9:30am on Day 1, and 9:30am, 12:30pm and 3:30pm on Day 2.

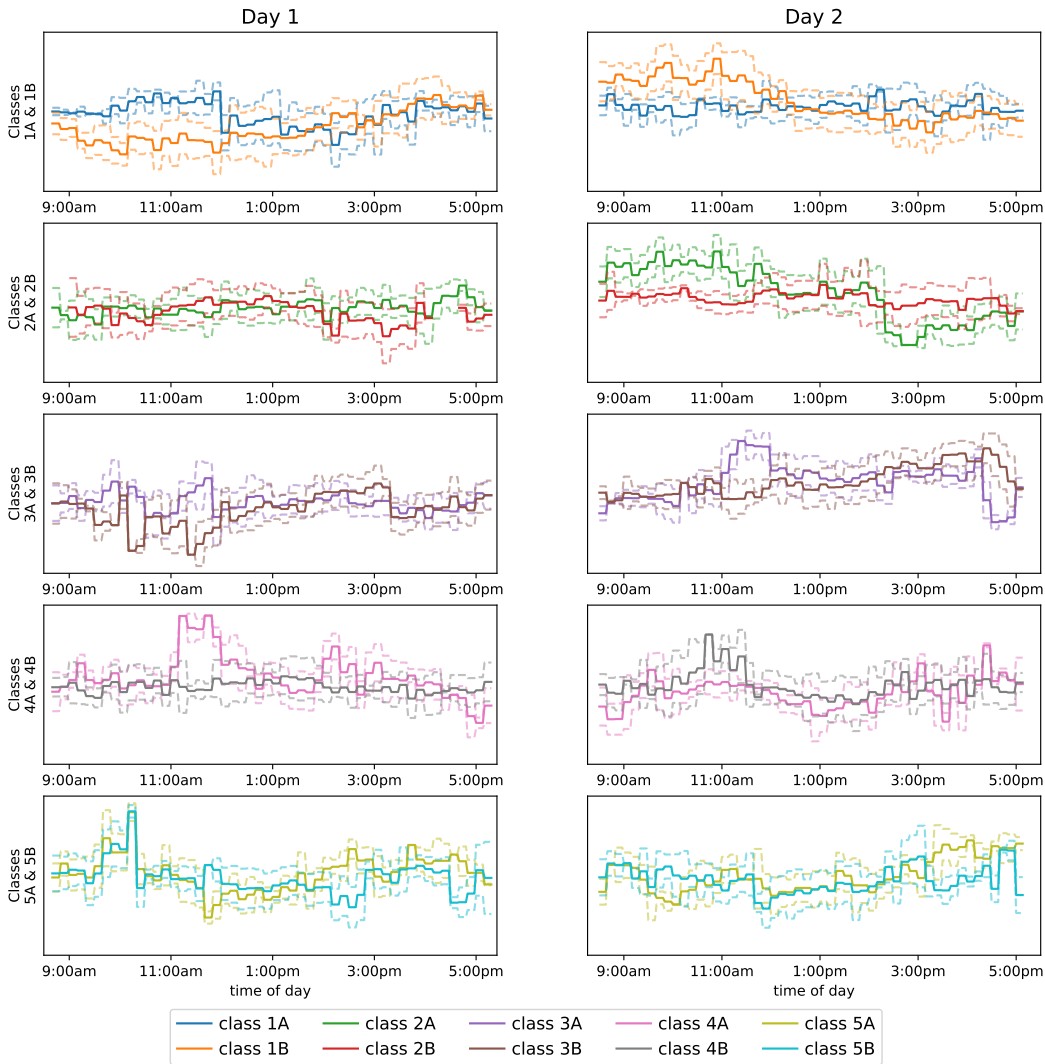

Figure 9: One-dimensional PCA visualisation of the 30-dimensional node representations for pairs of classes in the same year group, obtained using aligned spectral embedding. The solid lines show the average trajectory for each class, and the dashed line shows one standard deviation above and below.

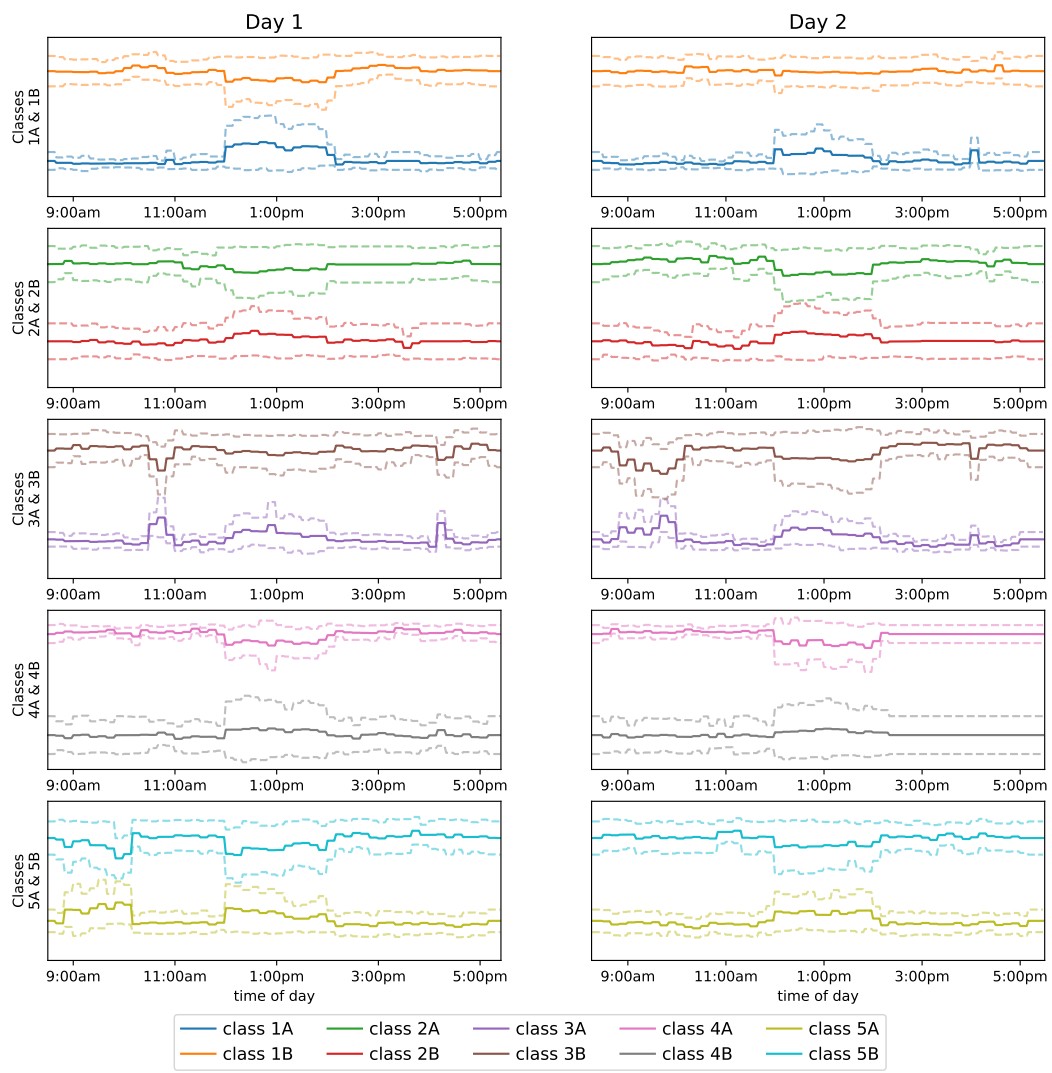

Figure 10: One-dimensional PCA visualisation of the 30-dimensional node representations for pairs of classes in the same year group, obtained using the Omnibus spectral embedding. The solid lines show the average trajectory for each class, and the dashed line shows one standard deviation above and below.

# B   Additional real data analysis

We provide further experiments and details of the Intensity Profile Projection analysis of the Lyon primary school dataset described in Section 5. As a comparison to the analysis in the paper, we apply Intensity Profile Projection to the data corresponding to each day of the study with a histogram intensity estimator, choosing a bin size of 10 minutes and computing 30-dimensional trajectories.

Figures 5 and 6 show the first two spherical coordinates [63] of the trajectories obtained using the histogram intensity estimator and the Epanechnikov kernel smoother, respectively. The six plots correspond to the morning, lunchtime and afternoon across both days.

Figure 7 (equivalent to Figure 3) visualises the trajectories of each pair of classes in each year group using PCA where we plot the average trajectory for each class, along with one standard deviation above and below. Since every trajectory using the histogram intensity estimator is piece-wise constant, so are the resulting PCA averages. The pairs of trajectories merge and split in a similar way to those obtained using the kernel smoother.

Figure 8 shows t-SNE visualisations of the node representations at a collection of times throughout the study. The plots are very similar to the equivalent Figure 4 for the kernel smoother with almost identical clusters of students before and after lunch, albeit placed differently by the t-SNE algorithm.

We apply the aligned spectral embedding method and the Omnibus spectral embedding method, described in Section 4.2 on the main text, to snapshots of the data. Figures 9 and 10 visualises the respective trajectories of each pair of classes in each year group using PCA where we plot the average trajectory for each class, along with one standard deviation above and below. These representations fail to capture the behaviour of the network and fail in the same ways as they did for the simulated data in Section 4.

# C   Visualisation

In this section, we give a short overview of the two dimension reduction techniques employed for visualisation in this paper.

For the trajectory visualisation in Figures 2 and 3, we use a principal component analysis which we extend to the dynamic setting by computing a projection using the leading eigenvectors of the *average* covariance matrix, which we apply to the (globally centered) trajectories. This has a similar flavour to our Intensity Profile Projection algorithm, and since we reduce dimension using a common projection, it gives a temporally coherent view of the trajectories.

The second visualisation technique we apply is t-SNE [64], using the Flt-SNE implementation [65], which we used to obtain Figure 4. This visualisation method is not naturally extended to dynamic data, so we initialise the algorithm using the aforementioned dynamic extension PCA, which results in the visualisations at different times being approximately aligned.

# D   Computational complexity

In this section, we given a brief discussion of the computational complexity of Algorithm 1. Suppose we use a kernel with finite support of width $2h$, then the expected time complexity of the kernel intensity estimate for a single edge at a single point in time is $O(h\lambda_{\max})$. Evaluation of $\widehat{\mathbf{\Lambda}}(t)$ is $O(n^2 h\lambda_{\max})$ and of the matrix (3) is $O(Bn^2 h\lambda_{\max})$. Consider the setting of Corollary 1, where $\lambda_{ij}(t) \asymp \rho$ for all $i, j, t$ and $L \asymp \rho L_0$ where $\mu, d, L_0 \asymp 1$ are fixed, our theory suggests choosing $h \asymp (n\rho)^{-1/3}$, and if we suppose $n\rho \asymp \log^3 n$, the sparsest regime our theory allows, then evaluation of (3) is $O(Bn\log^3 n)$, i.e. log-linear in $n$.

In practice the top singular vectors of (3) can be computed using the Augmented Implicitly Restarted Lanczos Bidiagonalization algorithm [47] implemented in the `irlba` package in R, or the `irlbpy` in Python. The time complexity of this algorithm has not been studied theoretically although in practice it can be incredibly fast. For example, the package author performs a simulated experiment in which they compute the first 2 singular vectors of a sparse 10M x 10M matrix with 1M non-zero entries which takes approximately 6 seconds on a computer with two Intel Xeon CPU E5-2650

processors (16 physical CPU cores) running at 2 GHz equipped with 128 GB of ECC DDR3 RAM (see https://bwlewis.github.io/irlba/comparison.html).

# E   Proof of Lemma 1

We begin by writing

$$
\underset{\mathbf{V}\in\mathcal{O}(n,d)}{\arg\min}\;\widehat{R}^2(\mathbf{V}) = \underset{\mathbf{V}\in\mathcal{O}(n,d)}{\arg\min}\int_0^T \widehat{r}_i^2(t;\mathbf{V})\,\mathrm{d}t
$$

$$
= \underset{\mathbf{V}\in\mathcal{O}(n,d)}{\arg\min}\int_0^T \sum_{i=1}^n \left\|\mathbf{V}\mathbf{V}^\top\widehat{\Lambda}_i(t) - \widehat{\Lambda}_i(t)\right\|_2^2\,\mathrm{d}t
$$

$$
= \underset{\mathbf{V}\in\mathcal{O}(n,d)}{\arg\min}\int_0^T \sum_{i=1}^n \left\|(\mathbf{I} - \mathbf{V}\mathbf{V}^\top)\widehat{\Lambda}_i(t)\right\|_2^2\,\mathrm{d}t,
$$

and since $(\mathbf{I} - \mathbf{V}\mathbf{V}^\top)$ is the projection onto the orthogonal complement of the columns space of $\mathbf{V}$, we have that

$$
\left\|\widehat{\Lambda}_i(t)\right\|_2^2 = \left\|\mathbf{V}\mathbf{V}^\top\widehat{\Lambda}_i(t)\right\|_2^2 + \left\|(\mathbf{I} - \mathbf{V}\mathbf{V}^\top)\widehat{\Lambda}_i(t)\right\|_2^2.
$$

Therefore, minimising $\widehat{R}^2(\mathbf{V})$ is equivalent to maximising

$$
\int_0^T \sum_{i=1}^n \left\|\mathbf{V}\mathbf{V}^\top\widehat{\Lambda}_i(t)\right\|_2^2\,\mathrm{d}t = \int_0^T \sum_{i=1}^n \left\|\mathbf{V}^\top\widehat{\Lambda}_i(t)\right\|_2^2
$$

where the equality holds due to the invariance of the Euclidean norm under orthogonal transformations. As a result, we have

$$
\underset{\mathbf{V}\in\mathcal{O}(n,d)}{\arg\min}\;\widehat{R}^2(\mathbf{V}) = \underset{\mathbf{V}\in\mathcal{O}(n,d)}{\arg\max}\int_0^T \sum_{i=1}^n \left\|\mathbf{V}^\top\widehat{\Lambda}_i(t)\right\|_2^2
$$

$$
= \underset{\mathbf{V}\in\mathcal{O}(n,d)}{\arg\max}\int_0^T \left\|\widehat{\mathbf{\Lambda}}(t)\mathbf{V}\right\|_F^2
$$

$$
= \underset{\mathbf{V}\in\mathcal{O}(n,d)}{\arg\max}\int_0^T \mathrm{tr}\left\{\mathbf{V}^\top\widehat{\mathbf{\Lambda}}^2(t)\mathbf{V}\right\}\,\mathrm{d}t
$$

$$
= \underset{\mathbf{V}\in\mathcal{O}(n,d)}{\arg\max}\;\mathrm{tr}\left\{\mathbf{V}^\top\left(\int_0^T \widehat{\mathbf{\Lambda}}^2(t)\,\mathrm{d}t\right)\mathbf{V}\right\}
$$

$$
= \underset{\mathbf{V}\in\mathcal{O}(n,d)}{\arg\max}\;\mathrm{tr}\left\{\mathbf{V}^\top\widehat{\mathbf{\Sigma}}\mathbf{V}\right\}
$$

$$
= \widehat{\mathbf{U}}
$$

where the final equality follows from the Courant-Fisher min-max theorem. This concludes the proof.

# F   Proof of Theorem 1

## F.1   Prerequisites

### F.1.1   Additional notation

In this proof, we use the notation $a_n \overset{\mathbb{P}}{\lesssim} b_n$ to mean $a_n \lesssim b_n$ with overwhelming probability.

### F.1.2   Symmetric dilation with change of basis "trick"

Symmetric dilation is a proof technique which allows statements about the eigenvectors of a symmetric matrix to be easily extended to hold for the singular vectors of a (potentially rectangular) asymmetric

matrix. Let $\mathbf{M}$ be an $n_1 \times n_2$ matrix with non-zero singular values $\{\sigma_i\}_{i=1}^r$ and corresponding orthonormal left singular vectors $\{u_i\}_{i=1}^r$ and right singular vectors $\{v_i\}_{i=1}^r$. Its *symmetric dilation* is the $n \times n$ matrix (with $n = n_1 + n_2$) constructed as

$$\mathcal{D}(\mathbf{M}) = \begin{pmatrix} \mathbf{0} & \mathbf{M} \\ \mathbf{M}^\top & \mathbf{0} \end{pmatrix}.$$

One can easily verify that $\mathcal{D}(\mathbf{M})$ has eigenvalues $\{\pm\sigma_i\}_{i=1}^r$ and eigenvectors $\{(u_i^\top, \pm v_i^\top)^\top\}_{i=1}^r$. We stack the first $d$ left and right singular vectors into matrices $\mathbf{U} \in \mathbb{R}^{n_1 \times d}$ and $\mathbf{V} \in \mathbb{R}^{n_2 \times d}$, and stack the first $2d$ eigenvectors of $\mathcal{D}(\mathbf{M})$ into a matrix

$$\bar{\mathbf{U}} = \frac{1}{\sqrt{2}} \begin{pmatrix} \mathbf{U} & \mathbf{U} \\ \mathbf{V} & -\mathbf{V} \end{pmatrix}.$$

We then have

$$\|\mathbf{U}\|_{2,\infty} \vee \|\mathbf{V}\|_{2,\infty} = \left\|\bar{\mathbf{U}}\right\|_{2,\infty}, \qquad \text{and} \qquad \|\mathbf{M}\|_2 = \|\mathcal{D}(\mathbf{M})\|_2.$$

While this standard construction is very useful when $n_1 \asymp n_2$, it can lead to suboptimal bounds when $n_2 \gg n_1$, or $n_1 \gg n_2$, due to an issue about incoherence, which was first raised in [13]. The *incoherence* of a subspace $\mathcal{U}_0$ spanned by the orthonormal columns of a matrix $\mathbf{U}_0 \in \mathbb{R}^{n_0 \times d}$ is

$$\mu(\mathbf{U}_0) = \sqrt{\frac{n_0}{d}} \|\mathbf{U}_0\|_{2,\infty}.$$

To obtain a good entrywise eigenvector bound under a signal-plus-noise matrix model it is typically necessary that $\mu(\bar{\mathbf{U}}) \asymp 1$. Observe that

$$\mu(\bar{\mathbf{U}}) = \sqrt{\frac{n_1 + n_2}{2d}} \|\bar{\mathbf{U}}\|_{2,\infty} = \sqrt{\frac{n_1 + n_2}{2d}} \left(\|\mathbf{U}\|_{2,\infty} \vee \|\mathbf{V}\|_{2,\infty}\right) = \sqrt{\frac{n_1 + n_2}{2n_1}}\mu(\mathbf{U}) + \sqrt{\frac{n_1 + n_2}{2n_2}}\mu(\mathbf{V}).$$

If $\mu(\mathbf{U}), \mu(\mathbf{V}) \asymp 1$ and $n_1 \asymp n_2$, then $\mu(\bar{\mathbf{U}}) \asymp 1$ and it is typically possible to obtain good bounds. However, when $n_2 \gg n_1$, we have $\mu(\bar{\mathbf{U}}) \gg 1$, and a good bound can typically not be obtained. The imbalance of $n_1$ and $n_2$ can cause similar issues when obtaining spectral norm bounds.

This issue can be overcome by changing to a basis which balances the contribution from its first $n_1$ and second $n_2$ elements of each column. Specifically, let $\pi_1 = \sqrt{2n_1/(n_1 + n_2)}$ and $\pi_2 = \sqrt{2n_2/(n_1 + n_2)}$, and consider the basis $\tilde{e}_1, \ldots, \tilde{e}_{n_1+n_2}$, such that

$$e_i = \begin{cases} \pi_1 \tilde{e}_i & \text{if } i \in \{1, \ldots, n_1\} \\ \pi_2 \tilde{e}_i & \text{if } i \in \{n_1 + 1, \ldots, n_1 + n_2\}, \end{cases}$$

where $\{e_i\}_{i=1}^{n_1+n_2}$ are the standard basis vectors in $\mathbb{R}^{n_0}$. Let $\|\!\|\cdot\|\!\|_\eta$ denote a norm with respect to the column basis $\{\tilde{e}_i\}_{i=1}^{n_1+n_2}$, then one can verify that

$$\|\!\|\mathcal{D}(\mathbf{M})\|\!\|_2 = \|\mathbf{M}\|, \qquad \text{and} \qquad \|\!\|\bar{\mathbf{U}}\|\!\|_{2,\infty} = \pi_1 \|\mathbf{U}\|_{2,\infty} \vee \pi_2 \|\mathbf{V}\|_{2,\infty}.$$

As a result, if $\tilde{\mu}(\mathbf{U}_0) = \sqrt{n_0/d}\|\!\|\mathbf{U}_0\|\!\|_{2,\infty}$, then

$$\tilde{\mu}(\bar{\mathbf{U}}) = \mu(\mathbf{U}) \vee \mu(\mathbf{V}),$$

regardless of the relative sizes of $n_1$ and $n_2$. We use this symmetric dilation with change-of-basis "trick" to apply some existing theorems for symmetric matrices to our setting.

### F.1.3 Concentration inequalities

In this section, we state a collection of lemmas which we will make use of throughout the proof. We begin with a tail bound for a Poisson random variable.

**Lemma 3.** *Let* $X \sim \text{Poisson}(\lambda)$. *Then*

$$\mathbb{P}(|X - \lambda| \geq t) \leq 2\exp\left(-\frac{t^2}{2(\lambda + t/3)}\right).$$

*For* $t \geq \lambda$,

$$\mathbb{P}(|X - \lambda| \geq t) \leq 2e^{-3t/8}.$$

The bound can be established by approximating the Poisson distribution with mean $\lambda$ as the sum of $k$ Bernoulli random variables with mean $\lambda/k$, applying Bernstein's inequality, and taking $k \to 0$.

Our next result is a concentration bound which adapts Lemma A.1 of [66] and can be proved using a vector version of the Bernstein inequality (Corollary 4.1 in [67]).

**Lemma 4.** *Let $X_i \sim \text{Poisson}(\lambda_i)$ independently for all $i = 1, \dots, n$, and suppose $\mathbf{Q} \in \mathbb{R}^{n \times d}$ is a deterministic matrix whose rows we denote $Q_i$. Let $\lambda_{\max} := \max_{i \in [n]} \lambda_i$, then with probability $1 - 28n^{-3}$*

$$\left\| \sum_{i=1}^{n} (X_i - \lambda_i) Q_i \right\|_2 \leq 3 \log^2 n \, \|\mathbf{Q}\|_{2,\infty} + \sqrt{6\lambda_{\max} \log n} \, \|\mathbf{Q}\|_{\mathrm{F}}.$$

Next, we state a concentration bound for the spectral norm of random matrices with independent entries which appears as Corollary 3.12 in [68]. The original statement of this lemma is for symmetric random matrices, although we general it to arbitrary random matrices using the symmetric dilation with change-of-basis trick described in Section F.1.2.

**Lemma 5** (Corollary 3.12 of [68]). *Let $\mathbf{X}$ be an $n_1 \times n_2$ matrix whose entries $x_{ij}$ are independent random variables which obey*

$$\mathbb{E}(x_{ij}) = 0, \quad \text{and} \quad |x_{ij}| \leq B, \quad i \in [n_1], j \in [n_2].$$

*Then there exists a universal constant $c > 0$ such that for any $t \geq 0$*

$$\mathbb{P}\left\{ \|\mathbf{X}\| \geq 4\sqrt{\nu} + t \right\} \leq n \exp\left( -\frac{t^2}{cB^2} \right).$$

*where*

$$\nu := \max\left\{ \pi_1 \max_{i \in [n_1]} \sum_{j=1}^{n_2} \mathbb{E}(x_{ij}^2), \pi_2 \max_{i \in [n_2]} \sum_{j=1}^{n_1} \mathbb{E}(x_{ji}^2) \right\}$$

*and $\pi_k = 2n_k/(n_1 + n_2)$.*

### F.1.4 Weyl's inequality and Wedin's sin$\Theta$ theorem

The next two lemmas are classical results matrix perturbation theory. Weyl's inequality shows that the singular values of a matrix are stable with respect to small perturbations.

**Lemma 6** (Weyl's inequality). *Let $\mathbf{M}, \mathbf{E}$ be $n_1 \times n_2$ real-valued matrices. Then for every $1 \leq i \leq (n_1 \wedge n_2)$, the $i$th largest singular value of $\mathbf{M}$ and $\mathbf{M} + \mathbf{E}$ obey*

$$|\sigma_i(\mathbf{M} + \mathbf{E}) - \sigma_i(\mathbf{M})| \leq \|\mathbf{E}\|_2.$$

One way to measure the distance between two subspaces $\mathcal{U}$ and $\widehat{\mathcal{U}}$ is via principal angles. Let $\mathbf{U}, \widehat{\mathbf{U}}$ be matrices whose orthonormal columns span $\mathcal{U}$ and $\widehat{\mathcal{U}}$ respectively, and let $\{\xi_i\}_{i=1}^{d}$ denote the singular values of $\mathbf{U}^\top \widehat{\mathbf{U}}$. Then the principal angles $\{\theta_i\}_{i=1}^{d}$ between $\mathcal{U}$ and $\widehat{\mathcal{U}}$ are defined by $\xi_i = \cos(\theta_i)$. Let $\sin\Theta(\mathbf{U}, \widehat{\mathbf{U}}) := \text{diag}(\sin\theta_1, \dots, \sin\theta_d)$. Another way to measure the distance between $\mathcal{U}$ and $\widehat{\mathcal{U}}$ is via the difference between the projection operators $\mathbf{U}\mathbf{U}^\top$ and $\widehat{\mathbf{U}}\widehat{\mathbf{U}}^\top$, and in fact, these two characterisations are equivalent. Specifically,

$$\left\| \sin\Theta(\mathbf{U}, \widehat{\mathbf{U}}) \right\|_2 \equiv \left\| \mathbf{U}\mathbf{U}^\top - \widehat{\mathbf{U}}\widehat{\mathbf{U}}^\top \right\|_2.$$

We will use this equivalence without mention throughout the proof. Wedin's sin$\Theta$ theorem shows that the singular vectors of a matrix are stable with respect to small perturbations.

**Lemma 7.** *Let $\mathbf{M}$ and $\widehat{\mathbf{M}} = \mathbf{M} + \mathbf{E}$ be two $n_1 \times n_2$ real-valued matrices, and denote by $\mathbf{U}, \widehat{\mathbf{U}}$ (respectively $\mathbf{V}, \widehat{\mathbf{V}}$) the matrices whose columns contain $d$ orthonormal left (respectively, right) singular vectors, corresponding to the $d$ largest singular values of $\mathbf{M}$ and $\widehat{\mathbf{M}}$. Let $\delta = \sigma_d(\mathbf{M}) - \sigma_{d+1}(\mathbf{M})$ and suppose that $\|\mathbf{E}\| < (1 - 1/\sqrt{2})\delta$, then*

$$\left\| \sin\Theta\left( \mathbf{U}, \widehat{\mathbf{U}} \right) \right\|_2 \vee \left\| \sin\Theta\left( \mathbf{V}, \widehat{\mathbf{V}} \right) \right\|_2 \leq \frac{2\left( \|\mathbf{E}^\top \mathbf{U}\|_2 \vee \|\mathbf{E}\mathbf{V}\|_2 \right)}{\delta} \leq \frac{2\|\mathbf{E}\|}{\delta}.$$

See [69] for a proof.

## F.2 Implications of Assumptions 1-4

We state here some inequalities involving the parameters of our problem which follow from Assumptions 1-4, and elementary linear algebra. We will use these facts throughout the proof without mention.

$$\sqrt{n\lambda_{\max}} \lesssim \delta \leq n\lambda_{\max}; \tag{6}$$

$$\delta \log n \gtrsim \kappa n\lambda_{\max}; \tag{7}$$

$$\kappa \lesssim \log n. \tag{8}$$

The inequality (6) holds since $\delta \leq \sigma_1^{1/2}(\boldsymbol{\Sigma}) \leq \sqrt{n}\|\boldsymbol{\Sigma}\|_{\max}^{1/2} \leq n\lambda_{\max}$, and

$$\delta \gtrsim \frac{\kappa n\lambda_{\max}}{\log(\delta/\sqrt{n\lambda_{max}})} \gtrsim \frac{n\lambda_{\max}}{\log n} \gtrsim \sqrt{n\lambda_{\max}\log n} \gtrsim \sqrt{n\lambda_{\max}}$$

where we invoked Assumption 4. (7) holds by noting that the previous bound implies $\log(\delta/\sqrt{n\lambda_{\max}}) \lesssim \log n$ and invoking Assumption 3. (8) follows from (6) since $\kappa \lesssim \delta \log n/n\lambda_{\max} \lesssim \log n$.

## F.3 Setup

We begin by defining $M$ equally spaced bins in $(0, 1]$,

$$B_1 := \left(0, \frac{1}{M}\right], \quad B_2 := \left(\frac{1}{M}, \frac{2}{M}\right], \ldots \quad, B_M := \left(\frac{M-1}{M}, 1\right],$$

and define the piecewise approximation of $\lambda_i(t)$,

$$\bar{\lambda}_i(t) = M \int_{B_m} \lambda_i(t)\mathrm{d}t, \qquad t \in B_m, \ m \in [M].$$

We then define $t_1, \ldots, t_m \in (0, 1]$ such that $\bar{\lambda}_{ij}(t) = \lambda_{ij}(t_m)$ for all $t \in B_m$, which exist by the continuity of $\lambda_{ij}(t)$, and define the piecewise constant approximation of $Y_i(t)$ as $\bar{Y}_i(t) = \mathbf{U}^\top \bar{\Lambda}_i(t)$. Our strategy to obtain the bound in Theorem 1 is to decompose it into bias and variance terms:

$$\max_{i,j\in[n]} \sup_{t\in\mathcal{T}} \left\|\mathbf{W}_1\widehat{Y}_i(t) - Y_i(t)\right\|_2 = \underbrace{\max_{i,j\in[n]} \sup_{t\in\mathcal{T}} \left\|\mathbf{W}_1\widehat{Y}_i(t) - \bar{Y}_i(t)\right\|_2}_{\text{variance}} + \underbrace{\max_{i,j\in[n]} \sup_{t\in\mathcal{T}} \left\|\mathbf{W}_2\bar{Y}_i(t) - Y_i(t)\right\|_2}_{\text{bias}}.$$

Section F.6 is dedicated to bounding the bias term, and the rest of this section is dedicated to bounding the variance term. Define the unfolding matrices $\widehat{\boldsymbol{\Lambda}}$ and $\boldsymbol{\Lambda}$ (without arguments) and their (thin) singular value decompositions as

$$\widehat{\boldsymbol{\Lambda}} := \left(\widehat{\boldsymbol{\Lambda}}(t_1) \ \cdots \ \widehat{\boldsymbol{\Lambda}}(t_M)\right) = \widehat{\mathbf{U}}\widehat{\mathbf{S}}\widehat{\mathbf{V}}^\top + \widehat{\mathbf{U}}_\perp\widehat{\mathbf{S}}_\perp\widehat{\mathbf{V}}_\perp^\top,$$

$$\bar{\boldsymbol{\Lambda}} := \left(\boldsymbol{\Lambda}(t_1) \ \cdots \ \boldsymbol{\Lambda}(t_M)\right) = \bar{\mathbf{U}}\bar{\mathbf{S}}\bar{\mathbf{V}}^\top + \bar{\mathbf{U}}_\perp\bar{\mathbf{S}}_\perp\bar{\mathbf{V}}_\perp^\top.$$

Then one has that for $t \in B_m$, $m \in [M]$,

$$\widehat{\mathbf{Y}}(t) := \widehat{\boldsymbol{\Lambda}}(t_m)\widehat{\mathbf{U}} = \widehat{\mathbf{V}}_m\widehat{\mathbf{S}}, \qquad \bar{\mathbf{Y}}(t) := \bar{\boldsymbol{\Lambda}}(t_m)\bar{\mathbf{U}} = \bar{\mathbf{V}}_m\widehat{\mathbf{S}}$$

where $\widehat{\mathbf{V}}_m, \bar{\mathbf{V}}_m$ denote the $m$th blocks of $\widehat{\mathbf{V}}$ and $\bar{\mathbf{V}}$ respectively. Therefore it follows that,

$$\max_{i,j\in[n]} \sup_{t\in\mathcal{T}} \left\|\mathbf{W}_1\widehat{Y}_i(t) - \bar{Y}_i(t)\right\|_2 = \left\|\widehat{\mathbf{V}}\widehat{\mathbf{S}}\mathbf{W}_1^\top - \bar{\mathbf{V}}\bar{\mathbf{S}}\right\|_{2,\infty}.$$

For ease of exposition, we drop the subscript 1 on $\mathbf{W}_1$ in this section. Our bound is based on the following decomposition of $\widehat{\mathbf{V}}\widehat{\mathbf{S}} - \bar{\mathbf{V}}\bar{\mathbf{S}}\mathbf{W}$.

**Proposition 1.** *We have the decomposition*

$$\widehat{\mathbf{V}}\widehat{\mathbf{S}} - \bar{\mathbf{V}}\bar{\mathbf{S}}\mathbf{W} = \bar{\mathbf{V}}(\bar{\mathbf{V}}^\top\widehat{\mathbf{V}}\widehat{\mathbf{S}} - \bar{\mathbf{S}}\mathbf{W}) \tag{9}$$

$$+ (\mathbf{I} - \bar{\mathbf{V}}\bar{\mathbf{V}}^\top)\bar{\boldsymbol{\Lambda}}^\top(\widehat{\mathbf{U}} - \bar{\mathbf{U}}\mathbf{W}) \tag{10}$$

$$+ (\mathbf{I} - \bar{\mathbf{V}}\bar{\mathbf{V}}^\top)(\widehat{\boldsymbol{\Lambda}} - \bar{\boldsymbol{\Lambda}})^\top\bar{\mathbf{U}}\mathbf{W} \tag{11}$$

$$+ (\mathbf{I} - \bar{\mathbf{V}}\bar{\mathbf{V}}^\top)(\widehat{\boldsymbol{\Lambda}} - \bar{\boldsymbol{\Lambda}})^\top(\widehat{\mathbf{U}} - \bar{\mathbf{U}}\mathbf{W}). \tag{12}$$

*Proof of Proposition 1.* We begin by adding and subtracting terms to obtain

$$\widehat{\mathbf{V}}\widehat{\mathbf{S}} - \bar{\mathbf{V}}\bar{\mathbf{S}}\mathbf{W} = \widehat{\mathbf{V}}\widehat{\mathbf{S}} - \bar{\mathbf{V}}\bar{\mathbf{V}}^\top\widehat{\mathbf{V}}\widehat{\mathbf{S}} + \underbrace{\bar{\mathbf{V}}(\bar{\mathbf{V}}^\top\widehat{\mathbf{V}}\widehat{\mathbf{S}} - \bar{\mathbf{S}}\mathbf{W})}_{(9)}.$$

Then, noting that $\widehat{\mathbf{V}}\widehat{\mathbf{S}} = \widehat{\mathbf{\Lambda}}^\top\widehat{\mathbf{U}}$ and $(\mathbf{I} - \bar{\mathbf{V}}\bar{\mathbf{V}}^\top)\bar{\mathbf{\Lambda}}^\top\bar{\mathbf{U}} = \mathbf{0}$, we have

$$
\begin{aligned}
\widehat{\mathbf{V}}\widehat{\mathbf{S}} - \bar{\mathbf{V}}\bar{\mathbf{V}}^\top\widehat{\mathbf{V}}\widehat{\mathbf{S}} &= \widehat{\mathbf{\Lambda}}^\top\widehat{\mathbf{U}} - \bar{\mathbf{V}}\bar{\mathbf{V}}^\top\widehat{\mathbf{\Lambda}}^\top\widehat{\mathbf{U}} \\
&= (\mathbf{I} - \bar{\mathbf{V}}\bar{\mathbf{V}}^\top)\widehat{\mathbf{\Lambda}}^\top\widehat{\mathbf{U}} \\
&= (\mathbf{I} - \bar{\mathbf{V}}\bar{\mathbf{V}}^\top)(\widehat{\mathbf{\Lambda}} - \bar{\mathbf{\Lambda}})^\top\widehat{\mathbf{U}} - (\mathbf{I} - \bar{\mathbf{V}}\bar{\mathbf{V}}^\top)\bar{\mathbf{\Lambda}}^\top\widehat{\mathbf{U}} \\
&= (\mathbf{I} - \bar{\mathbf{V}}\bar{\mathbf{V}}^\top)(\widehat{\mathbf{\Lambda}} - \bar{\mathbf{\Lambda}})^\top\widehat{\mathbf{U}} - \underbrace{(\mathbf{I} - \bar{\mathbf{V}}\bar{\mathbf{V}}^\top)\bar{\mathbf{\Lambda}}^\top(\widehat{\mathbf{U}} - \bar{\mathbf{U}}\mathbf{W})}_{(10)}.
\end{aligned}
$$

Next, we decompose $(\mathbf{I} - \bar{\mathbf{V}}\bar{\mathbf{V}}^\top)(\widehat{\mathbf{\Lambda}} - \bar{\mathbf{\Lambda}})^\top\widehat{\mathbf{U}}$ by adding and subtracting terms to obtain

$$(\mathbf{I} - \bar{\mathbf{V}}\bar{\mathbf{V}}^\top)(\widehat{\mathbf{\Lambda}} - \bar{\mathbf{\Lambda}})^\top\widehat{\mathbf{U}} = \underbrace{(\mathbf{I} - \bar{\mathbf{V}}\bar{\mathbf{V}}^\top)(\widehat{\mathbf{\Lambda}} - \bar{\mathbf{\Lambda}})^\top\bar{\mathbf{U}}\mathbf{W}}_{(11)} + \underbrace{(\mathbf{I} - \bar{\mathbf{V}}\bar{\mathbf{V}}^\top)(\widehat{\mathbf{\Lambda}} - \bar{\mathbf{\Lambda}})^\top(\widehat{\mathbf{U}} - \mathbf{U}\mathbf{W})}_{(12)}.$$

$\square$

## F.4   Technical propositions

We now outline a series of technical propositions which we require to bound terms (9)-(12) which we prove in Section G.

Our first proposition is a 1-norm and spectral norm bound for $\widehat{\mathbf{\Lambda}}$.

**Proposition 2.** *The bounds*

$$\left\|\widehat{\mathbf{\Lambda}} - \bar{\mathbf{\Lambda}}\right\|_1 \lesssim \sqrt{Mn\lambda_{\max}\log n}, \qquad \left\|\widehat{\mathbf{\Lambda}} - \bar{\mathbf{\Lambda}}\right\|_2 \lesssim \sqrt{Mn\lambda_{\max}}$$

*hold with overwhelming probability.*

The spectral norm bound is obtained using Lemma 5, and the 1-norm bound is obtained via an application of the classical Bernstein inequality. The next proposition provides control on the singular values of $\widehat{\mathbf{\Lambda}}$.

**Proposition 3.** *Let $\sigma_i(\cdot)$ denote the $i$th ordered singular value of a matrix. The singular values of $\widehat{\mathbf{\Lambda}}$ satisfy*

$$\sqrt{M\sigma_d(\mathbf{\Sigma})} \lesssim \sigma_d(\widehat{\mathbf{\Lambda}}) \leq \sigma_1(\widehat{\mathbf{\Lambda}}) \lesssim \sqrt{M\sigma_1(\mathbf{\Sigma})}.$$

The result is obtained using Weyl's inequality. The next proposition provides control of the spectral norm of $\mathbf{Q}^\top(\widehat{\mathbf{\Lambda}} - \bar{\mathbf{\Lambda}})\mathbf{R}$, where $\mathbf{Q}, \mathbf{R}$ are conformable, deterministic unit-norm matrices.

**Proposition 4.** *For conformable, deterministic unit-norm matrices $\mathbf{Q}, \mathbf{R}$, the bound*

$$\left\|\mathbf{Q}^\top(\widehat{\mathbf{\Lambda}} - \bar{\mathbf{\Lambda}})\mathbf{R}\right\|_2 \lesssim M\log^{3/2} n \tag{13}$$

*holds with overwhelming probability.*

The proof of Proposition 4 employs a classical $\varepsilon$-net argument to the spectral norm of an appropriately constructed symmetric dilation matrix.

The next proposition states that both the matrices $\bar{\mathbf{U}}^\top\widehat{\mathbf{U}}$ and $\bar{\mathbf{V}}^\top\widehat{\mathbf{V}}$ are well approximated by a common orthogonal matrix.

**Proposition 5.** *There exists an orthogonal matrix $\mathbf{W}$ such that*

$$\left\|\bar{\mathbf{U}}^\top\widehat{\mathbf{U}} - \mathbf{W}\right\|_2 \lesssim \frac{\sqrt{n\lambda_{\max}}}{\delta}, \qquad \left\|\bar{\mathbf{V}}^\top\widehat{\mathbf{V}} - \mathbf{W}\right\|_2 \lesssim \frac{\sqrt{n\lambda_{\max}}}{\delta}$$

*hold with overwhelming probability.*

To prove Proposition 5, we empoy the Wedin $\sin\Theta$ theorem to obtain a bound on $\|\bar{\mathbf{U}}^\top\widehat{\mathbf{U}} - \mathbf{W}\|_2$. We then obtain a bound on $\|\bar{\mathbf{U}}^\top\widehat{\mathbf{U}} - \bar{\mathbf{V}}^\top\widehat{\mathbf{V}}\|_2$, and combine these bounds to establish the proposition.

The next technical tool we require is the ability to "swap" $\mathbf{W}, \bar{\mathbf{S}}$ and $\widehat{\mathbf{S}}$.

**Proposition 6.** *The bound*

$$\left\|\mathbf{W}\widehat{\mathbf{S}} - \bar{\mathbf{S}}\mathbf{W}\right\|_2 \lesssim M\log^{3/2} n$$

*holds with overwhelming probability.*

This result follows by applying the previous propositions to an appropriately constructed decomposition.

Part of the challenge of obtaining a good bound on the term (12) is that $(\tilde{\mathbf{A}} - \tilde{\mathbf{\Lambda}})$ and $(\widehat{\mathbf{U}} - \bar{\mathbf{U}}\mathbf{W})$ are dependent, and this dependence must be decoupled in order to apply the standard suite of matrix perturbation tools. For $m = 1, \dots, n$, let

$$\mathcal{N}_m = \{(i,j) : i = m \text{ or } j \in \{m + (\ell-1)n, \ell \in [M]\}\}$$

and construct the auxiliary matrices $\widehat{\mathbf{\Lambda}}^{(1)}, \dots, \widehat{\mathbf{\Lambda}}^{(n)}$ defined by

$$\widehat{\mathbf{\Lambda}}_{ij}^{(m)} = \begin{cases} \widehat{\mathbf{\Lambda}}_{ij} & \text{if } (i,j) \notin \mathcal{N}_m, \\ \bar{\mathbf{\Lambda}}_{ij} & \text{if } (i,j) \in \mathcal{N}_m. \end{cases} \tag{14}$$

In words, $\widehat{\mathbf{\Lambda}}_{ij}^{(m)}$ is the matrix obtained by replacing the $m$th row and columns of each of its blocks with its expectation. In this way, the $m$th row of $(\widehat{\mathbf{\Lambda}} - \bar{\mathbf{\Lambda}})$ and $\widehat{\mathbf{\Lambda}}^{(m)}$ are independent. Let $\widehat{\mathbf{U}}^{(m)}$ denote the matrix of leading left singular values of $\widehat{\mathbf{\Lambda}}^{(m)}$.

We apply a result due to [11], which provides $\ell_{2,\infty}$ control of $\|\widehat{\mathbf{U}}\|_{2,\infty}, \|\widehat{\mathbf{U}}^{(m)}\|_{2,\infty}$, and $\|\widehat{\mathbf{U}}^{(m)}\mathbf{W}^{(m)} - \mathbf{U}\|_{2,\infty}$.

**Proposition 7.** *The bounds*

$$\left\|\widehat{\mathbf{U}}\right\|_{2,\infty}, \ \left\|\widehat{\mathbf{U}}^{(m)}\right\|_{2,\infty}, \ \left\|\widehat{\mathbf{U}}^{(m)}\mathbf{W}^{(m)} - \mathbf{U}\right\|_{2,\infty} \lesssim \frac{\mu\lambda_{\max}\sqrt{dn}\log n}{\delta}$$

*hold with overwhelming probability.*

In addition, we require control on the spectral norm difference between the projection matrices $\widehat{\mathbf{U}}^{(m)}(\widehat{\mathbf{U}}^{(m)})^\top$ and the projection matrices $\bar{\mathbf{U}}\bar{\mathbf{U}}^\top$ and $\widehat{\mathbf{U}}\widehat{\mathbf{U}}^\top$, which is provided in the following proposition.

**Proposition 8.** *The bounds*

$$\left\|\widehat{\mathbf{U}}^{(m)}(\widehat{\mathbf{U}}^{(m)})^\top - \bar{\mathbf{U}}\bar{\mathbf{U}}^\top\right\|_2 \lesssim \frac{n\lambda_{\max}}{\delta}, \tag{15}$$

$$\left\|\widehat{\mathbf{U}}^{(m)}(\widehat{\mathbf{U}}^{(m)})^\top - \widehat{\mathbf{U}}\widehat{\mathbf{U}}^\top\right\|_2 \lesssim \frac{\mu\lambda_{\max}^{3/2}\sqrt{dn}\log^{3/2} n}{\delta^2} \tag{16}$$

*hold with overwhelming probability.*

The proof of Proposition 8 requires a delicate "leave-one-out"–style argument.

### F.5  Bounding terms (9)-(12)

Firstly observe that

$$\|\bar{\mathbf{V}}\|_{2,\infty} = \|\bar{\mathbf{\Lambda}}^\top\bar{\mathbf{U}}\bar{\mathbf{S}}^{-1}\|_{2,\infty} \leq \|\bar{\mathbf{\Lambda}}^\top\|_\infty \|\bar{\mathbf{U}}\|_{2,\infty} \|\bar{\mathbf{S}}^{-1}\|_2 \leq \frac{\sqrt{nd}\lambda_{\max}\mu}{\sqrt{M\sigma_d(\mathbf{\Sigma})}}$$

and therefore term (9) can be bounded as

$$\left\|\bar{\mathbf{V}}(\bar{\mathbf{V}}^\top\widehat{\mathbf{V}}\widehat{\mathbf{S}}-\bar{\mathbf{S}}\mathbf{W})\right\|_{2,\infty} \le \left\|\bar{\mathbf{V}}\right\|_{2,\infty}\left(\left\|\bar{\mathbf{V}}^\top\widehat{\mathbf{V}}-\mathbf{W}\right\|_2\left\|\widehat{\mathbf{S}}\right\| + \left\|\mathbf{W}\widehat{\mathbf{S}}-\bar{\mathbf{S}}\mathbf{W}\right\|_2\right)$$

$$\overset{\mathbb{P}}{\lesssim} \frac{\sqrt{nd}\lambda_{\max}\mu}{\sqrt{M}\sigma_d(\boldsymbol{\Sigma})}\left(\frac{\sqrt{n\lambda_{\max}}}{\delta}\cdot\sqrt{M\sigma_1(\boldsymbol{\Sigma})}+M\log^{3/2}n\right)$$

$$\lesssim \frac{n\sqrt{M}d\lambda_{\max}^{3/2}\mu\kappa}{\delta}$$

$$\lesssim \mu\sqrt{M\lambda_{\max}}d\log n.$$

where the third inequality follows from Assumption 4 that $\sqrt{M}\log^{3/2}n \lesssim n\lambda_{\max}$, and the definition $\kappa := \sqrt{\sigma_1(\boldsymbol{\Sigma})/\sigma_d(\boldsymbol{\Sigma})}$, and the fourth inequality follows from Assumption 3 that $\delta\log n \ge \delta\log(\delta/\sqrt{n\lambda_{\max}}) \gtrsim \kappa n\lambda_{max}$.

To bound (10), we first apply Wedin's $\sin\Theta$ theorem to obtain

$$\left\|\widehat{\mathbf{U}}-\bar{\mathbf{U}}\mathbf{W}\right\|_2 = \left\|\sin\Theta(\widehat{\mathbf{U}},\bar{\mathbf{U}})\right\|_2 \le \frac{\left\|\widehat{\boldsymbol{\Lambda}}-\bar{\boldsymbol{\Lambda}}\right\|}{\sigma_d(\bar{\boldsymbol{\Lambda}})-\sigma_{d+1}(\bar{\boldsymbol{\Lambda}})} \overset{\mathbb{P}}{\lesssim} \frac{\sqrt{Mn\lambda_{\max}}}{\sqrt{M}\delta} = \frac{\sqrt{n\lambda_{\max}}}{\delta}$$

Then, we use Assumption 2 to obtain the bound

$$\left\|(\mathbf{I}-\bar{\mathbf{V}}\bar{\mathbf{V}}^\top)\bar{\boldsymbol{\Lambda}}^\top\right\|_{2,\infty} = \left\|\bar{\boldsymbol{\Lambda}}^\top(\mathbf{I}-\bar{\mathbf{U}}\bar{\mathbf{U}})\right\|_{2,\infty} \lesssim \max_{i\in[n]}\sup_{t\in\mathcal{T}}r_i(t) \lesssim \sqrt{\frac{d}{n}}\mu\delta\log^{5/2}n.$$

Putting these two bounds together, we bound (10) as

$$\left\|(\mathbf{I}-\bar{\mathbf{V}}\bar{\mathbf{V}}^\top)\bar{\boldsymbol{\Lambda}}^\top(\widehat{\mathbf{U}}-\bar{\mathbf{U}}\mathbf{W})\right\|_{2,\infty} \le \left\|(\mathbf{I}-\bar{\mathbf{V}}\bar{\mathbf{V}}^\top)\bar{\boldsymbol{\Lambda}}^\top\right\|_{2,\infty}\left\|\widehat{\mathbf{U}}-\bar{\mathbf{U}}\mathbf{W}\right\|_2 \overset{\mathbb{P}}{\lesssim} \mu\sqrt{d\lambda_{\max}}\log^{5/2}n$$

To bound term (11), we set $\mathbf{E}=\widehat{\boldsymbol{\Lambda}}-\bar{\boldsymbol{\Lambda}}$ and note that each column of $M^{-1}\mathbf{E}$ contains independent Poisson random variables with means no greater that $M^{-1}\lambda_{\max}$. We will use Lemma 4 to bound the rows $\mathbf{E}\bar{\mathbf{U}}$ as

$$[\mathbf{E}^\top\bar{\mathbf{U}}]_i = \sum_{j=1}^n e_{ji}\bar{U}_j \overset{\mathbb{P}}{\lesssim} M\log^2 n\left\|\bar{\mathbf{U}}\right\|_{2,\infty} + \sqrt{M\lambda_{\max}\log n}\left\|\bar{\mathbf{U}}\right\|_{\mathrm{F}}$$

$$\lesssim \left(M\log^2 n + \sqrt{Mn\lambda_{\max}\log n}\right)\left\|\bar{\mathbf{U}}\right\|_{2,\infty}$$

$$\lesssim \sqrt{M\lambda_{\max}n\log n}\left\|\bar{\mathbf{U}}\right\|_{2,\infty}$$

$$\lesssim \mu\sqrt{M\lambda_{\max}d\log n}.$$

where the third inequality uses Assumption 4 and a union bound over $i=1,\dots,n$. Therefore, we have

$$\|(\widehat{\boldsymbol{\Lambda}}-\bar{\boldsymbol{\Lambda}})^\top\bar{\mathbf{U}}\|_{2,\infty} \overset{\mathbb{P}}{\lesssim} \sqrt{M\lambda_{\max}d\log n}. \tag{17}$$

Noting that $\left\|\mathbf{I}-\bar{\mathbf{V}}\bar{\mathbf{V}}^\top\right\|_\infty \le \left\|\mathbf{I}-\bar{\mathbf{V}}\bar{\mathbf{V}}^\top\right\|_2 \lesssim 1$, we bound (11) as

$$\left\|(\mathbf{I}-\bar{\mathbf{V}}\bar{\mathbf{V}}^\top)(\widehat{\boldsymbol{\Lambda}}-\bar{\boldsymbol{\Lambda}})^\top\bar{\mathbf{U}}\mathbf{W}\right\|_{2,\infty} \lesssim \left\|\mathbf{I}-\bar{\mathbf{V}}\bar{\mathbf{V}}\right\|_\infty\left\|\left(\widehat{\boldsymbol{\Lambda}}-\bar{\boldsymbol{\Lambda}}\right)^\top\bar{\mathbf{U}}\right\|_{2,\infty} \overset{\mathbb{P}}{\lesssim} \mu\sqrt{M\lambda_{\max}d\log n}.$$

Finally, we bound term (12). Let $\widehat{\boldsymbol{\Lambda}}^{(1)},\dots,\widehat{\boldsymbol{\Lambda}}^{(n)}$ denote the auxiliary matrices described in (14), and let $\widehat{\mathbf{U}}^{(m)}$ denote the matrix of leading left singular values of $\widehat{\boldsymbol{\Lambda}}^{(m)}$. We can then decompose the Euclidean norm of $(\widehat{\boldsymbol{\Lambda}}-\bar{\boldsymbol{\Lambda}})_{\cdot,m}^\top(\widehat{\mathbf{U}}-\mathbf{U}\mathbf{W})$ as

$$\left\|(\widehat{\boldsymbol{\Lambda}}-\bar{\boldsymbol{\Lambda}})_{\cdot,m}^\top(\widehat{\mathbf{U}}-\bar{\mathbf{U}}\mathbf{W})\right\|_2 \le \left\|(\widehat{\boldsymbol{\Lambda}}-\bar{\boldsymbol{\Lambda}})_{\cdot,m}^\top\widehat{\mathbf{U}}(\mathbf{W}-\widehat{\mathbf{U}}^\top\bar{\mathbf{U}})\right\|_2 \tag{18}$$

$$+ \left\|(\widehat{\boldsymbol{\Lambda}}-\bar{\boldsymbol{\Lambda}})_{\cdot,m}^\top(\widehat{\mathbf{U}}\widehat{\mathbf{U}}^\top\bar{\mathbf{U}}-\widehat{\mathbf{U}}^{(m)}(\widehat{\mathbf{U}}^{(m)})^\top\bar{\mathbf{U}})\right\|_2 \tag{19}$$

$$+ \left\|(\widehat{\boldsymbol{\Lambda}}-\bar{\boldsymbol{\Lambda}})_{\cdot,m}^\top(\widehat{\mathbf{U}}^{(m)}(\widehat{\mathbf{U}}^{(m)})^\top\bar{\mathbf{U}}-\bar{\mathbf{U}})\right\|_2. \tag{20}$$

The first term (18) is bounded as

$$
\left\|(\widehat{\boldsymbol{\Lambda}} - \bar{\boldsymbol{\Lambda}})_{\cdot,m}^{\top}\widehat{\mathbf{U}}(\mathbf{W} - \widehat{\mathbf{U}}^{\top}\bar{\mathbf{U}})\right\|_2 \leq \left\|\widehat{\boldsymbol{\Lambda}} - \bar{\boldsymbol{\Lambda}}\right\|_1 \left\|\widehat{\mathbf{U}}\right\|_{2,\infty} \left\|\mathbf{W} - \widehat{\mathbf{U}}^{\top}\bar{\mathbf{U}}\right\|_2
$$

$$
\overset{\mathbb{P}}{\lesssim} \sqrt{Mn\lambda_{max}\log n} \cdot \frac{\mu\lambda_{\max}\sqrt{dn}\log n}{\delta} \cdot \frac{\sqrt{n\lambda_{\max}}}{\delta}
$$

$$
= \frac{\sqrt{M}n^{3/2}\lambda_{\max}^2\mu\sqrt{d}\log^{3/2}n}{\delta^2}
$$

$$
\lesssim \mu\sqrt{M\lambda_{\max}d}\log^{5/2}n.
$$

To bound the second term (19), we employ Proposition 8 to obtain

$$
\left\|(\widehat{\boldsymbol{\Lambda}} - \bar{\boldsymbol{\Lambda}})_{\cdot,m}^{\top}(\widehat{\mathbf{U}}\widehat{\mathbf{U}}^{\top}\bar{\mathbf{U}} - \widehat{\mathbf{U}}^{(m)}(\widehat{\mathbf{U}}^{(m)})^{\top}\bar{\mathbf{U}})\right\|_2 \leq \left\|\widehat{\boldsymbol{\Lambda}} - \bar{\boldsymbol{\Lambda}}\right\|_2 \left\|\widehat{\mathbf{U}}\widehat{\mathbf{U}}^{\top} - \widehat{\mathbf{U}}^{(m)}(\widehat{\mathbf{U}}^{(m)})^{\top}\right\|_2
$$

$$
\overset{\mathbb{P}}{\lesssim} \sqrt{Mn\lambda_{\max}} \cdot \frac{\mu\lambda_{\max}^{3/2}\sqrt{dn}\log^{3/2}n}{\delta^2}
$$

$$
= \frac{\sqrt{M}\mu\lambda_{\max}^2\sqrt{d}n^{3/2}\log^{3/2}n}{\delta^2}
$$

$$
\lesssim \mu\sqrt{M\lambda_{\max}d}\log^{5/2}n
$$

We now set about bounding the third term (20). Let $\boldsymbol{\Omega}_1\boldsymbol{\Xi}\boldsymbol{\Omega}_2^{\top}$ denote a singular value decomposition of $(\widehat{\mathbf{U}}^{(m)})^{\top}\bar{\mathbf{U}}$, and set $\mathbf{W}^{(m)} := \boldsymbol{\Omega}_1\boldsymbol{\Omega}_2^{\top}$. Let $\theta_i^{(m)}$ denote the principal angles between the column spaces of $\widehat{\mathbf{U}}^{(m)}$ and $\bar{\mathbf{U}}$ defined by $\xi_i^{(m)} = \cos(\theta_i^{(m)})$, where $\xi_i^{(m)}$ are the singular values of $(\widehat{\mathbf{U}}^{(m)})^{\top}\bar{\mathbf{U}}$. We invoke Wedin's theorem to show that

$$
\left\|\mathbf{W}^{(m)} - \left(\widehat{\mathbf{U}}^{(m)}\right)^{\top}\bar{\mathbf{U}}\right\|_2 = \|\mathbf{I} - \boldsymbol{\Xi}\|_2 = \max_{i\in[d]}(1 - \xi_i^{(m)}) = \max_{i\in[d]}(1 - \cos\theta_i^{(m)})
$$

$$
\leq \max_{i\in[d]}(1 - \cos^2\theta_i^{(m)}) = \max_{i\in[d]}\sin^2\theta_i^{(m)} \lesssim \frac{\left\|\widehat{\boldsymbol{\Lambda}}^{(m)} - \bar{\boldsymbol{\Lambda}}\right\|_2^2}{(\sigma_d(\bar{\boldsymbol{\Lambda}}) - \sigma_{d+1}(\widehat{\boldsymbol{\Lambda}}))^2} \overset{\mathbb{P}}{\lesssim} \frac{Mn\lambda_{\max}}{M\delta^2} = \frac{n\lambda_{\max}}{\delta^2} \lesssim 1
$$

We define $\mathbf{H}^{(m)} := \widehat{\mathbf{U}}^{(m)}(\widehat{\mathbf{U}}^{(m)})^{\top} - \mathbf{U}\mathbf{U}^{\top}$ and note that $\mathbf{H}^{(m)}$ is independent of $(\widehat{\boldsymbol{\Lambda}} - \bar{\boldsymbol{\Lambda}})_{m,\cdot}$ and that

$$
\left\|\mathbf{H}^{(m)}\right\|_{2,\infty} \leq \left\|\widehat{\mathbf{U}}^{(m)}\mathbf{W}^{(m)} - \bar{\mathbf{U}}\right\|_{2,\infty} + \left\|\widehat{\mathbf{U}}^{(m)}\right\|_{2,\infty}\left\|(\widehat{\mathbf{U}}^{(m)})^{\top}\bar{\mathbf{U}} - \mathbf{W}^{(m)}\right\|_2
$$

$$
\lesssim \left\|\widehat{\mathbf{U}}^{(m)}\mathbf{W}^{(m)} - \bar{\mathbf{U}}\right\|_{2,\infty} + \left\|\widehat{\mathbf{U}}^{(m)}\right\|_{2,\infty}
$$

$$
\overset{\mathbb{P}}{\lesssim} \frac{\mu\lambda_{\max}\sqrt{dn}\log n}{\delta}
$$

Then, using Lemma 4 we have that

$$
\left\|(\widetilde{\mathbf{A}} - \widetilde{\boldsymbol{\Lambda}})_{\cdot,m}^{\top}\mathbf{H}^{(m)}\right\|_2 \overset{\mathbb{P}}{\lesssim} M\log^2 n\left\|\mathbf{H}^{(m)}\right\|_{2,\infty} + \sqrt{\lambda_{\max}\log n}\left\|\mathbf{H}^{(m)}\right\|_{\mathrm{F}}
$$

$$
\overset{\mathbb{P}}{\lesssim} \sqrt{M\log n\lambda_{\max}}\left\|\mathbf{H}^{(m)}\right\|_{2,\infty} + \sqrt{\lambda_{\max}d\log n}\left\|\mathbf{H}^{(m)}\right\|_2
$$

$$
\overset{\mathbb{P}}{\lesssim} \sqrt{M\log n\lambda_{\max}} \cdot \frac{\mu\lambda_{\max}\sqrt{dn}\log n}{\delta} + \sqrt{\lambda_{\max}d\log n}\frac{n\lambda_{\max}}{\delta}
$$

$$
\leq \frac{\sqrt{M}\mu n\lambda_{\max}^{3/2}\sqrt{d}\log^{3/2}n}{\delta}
$$

$$
\lesssim \mu\sqrt{Md\lambda_{\max}}\log^{5/2}n.
$$

Combining these bounds and taking a union bound over $m \in [n]$, we have

$$\left\| (\widehat{\boldsymbol{\Lambda}} - \bar{\boldsymbol{\Lambda}})^\top (\widehat{\mathbf{U}} - \bar{\mathbf{U}}\mathbf{W}) \right\|_{2,\infty} \overset{\mathbb{P}}{\lesssim} \mu \sqrt{Md\lambda_{\max}} \log^{5/2} n,$$

and the term (12) is bounded as

$$\left\| (\mathbf{I} - \bar{\mathbf{V}}\bar{\mathbf{V}}^\top)(\widehat{\boldsymbol{\Lambda}} - \bar{\boldsymbol{\Lambda}})^\top (\widehat{\mathbf{U}} - \bar{\mathbf{U}}\mathbf{W}) \right\|_{2,\infty} \leq \left\| \mathbf{I} - \bar{\mathbf{V}}\bar{\mathbf{V}} \right\|_\infty \left\| (\widehat{\boldsymbol{\Lambda}} - \bar{\boldsymbol{\Lambda}})^\top (\widehat{\mathbf{U}} - \bar{\mathbf{U}}\mathbf{W}) \right\|_{2,\infty}$$

$$\overset{\mathbb{P}}{\lesssim} \mu \sqrt{M\lambda_{\max} d} \log^{5/2} n.$$

Combining the bounds on (9)-(12), we have

$$\max_{i,j \in [n]} \sup_{t \in \mathcal{T}} \left\| \mathbf{W}_1 \widehat{Y}_i(t) - \bar{Y}_i(t) \right\|_2 = \left\| \widehat{\mathbf{V}}\widehat{\mathbf{S}}\mathbf{W}_1^\top - \bar{\mathbf{V}}\bar{\mathbf{S}} \right\|_{2,\infty} \overset{\mathbb{P}}{\lesssim} \mu \sqrt{Md\lambda_{\max}} \log^{5/2} n,$$

which completes the proof.

### F.6 Controlling the bias term

#### F.6.1 Edge level bias

We begin by studying the edge-level bias of the histogram intensity estimator. Let $\rho_{ij}(t) = \int_0^t \lambda_{ij}(s) \, ds$ denote the cumulative intensity of edge $i,j$. Now we have, for $t \in B_\ell$,

$$
\begin{aligned}
\bar{\lambda}_{ij}(t) &= M \int_{B_\ell} \lambda_{ij}(s) \, dt \\
&= M \left\{ \rho_{ij}\left(\frac{\ell}{M}\right) - \rho_{ij}\left(\frac{\ell-1}{M}\right) \right\} \\
&= \frac{\rho_{ij}\left(\frac{\ell}{M}\right) - \rho_{ij}\left(\frac{\ell-1}{M}\right)}{\frac{\ell}{M} - \frac{\ell-1}{M}} \\
&= \lambda_{ij}(t^\star)
\end{aligned}
$$

for some $t^\star \in B^\ell$, which follows by an application of the mean value theorem, where $\rho'(t) = \lambda_{ij}(t)$. We then apply the $L$-Lipschitz continuity of $\lambda_{ij}(t)$ to obtain

$$
\begin{aligned}
\bar{\lambda}_{ij}(t) - \lambda_{ij}(t)| = |\lambda_{ij}(t^\star) - \lambda_{ij}(t)| \\
\leq L \cdot |t^\star - t| \\
\leq \frac{L}{M}.
\end{aligned}
$$

#### F.6.2 A subspace perturbation bound

Define the operator $\mathcal{A} : (\mathcal{T} \to \mathbb{R}^n) \to \mathbb{R}^n$ by

$$\mathcal{A}v(\cdot) = \int_{\mathcal{T}} \boldsymbol{\Lambda}(t)v(t) \, dt$$

and define the operator $\mathcal{A}^\star : \mathbb{R}^n \to (\mathcal{T} \to \mathbb{R}^n)$ by

$$\mathcal{A}^\star u = \boldsymbol{\Lambda}(\cdot)u.$$

Then $\boldsymbol{\Sigma} \equiv \mathcal{A}\mathcal{A}^\star$ since

$$\mathcal{A}\mathcal{A}^\star u = \mathcal{A}\left(\boldsymbol{\Lambda}(\cdot)u\right) = \int_{\mathcal{T}} \boldsymbol{\Lambda}^2(t)u \, dt = \boldsymbol{\Sigma}u.$$

Denote its eigenvalues $\sigma_1^2, \ldots, \sigma_n^2$, and its corresponding orthonormal eigenvectors $u_1, \ldots, u_n$, and define $v_i(\cdot) = \boldsymbol{\Lambda}(\cdot)u_i/\xi_i$ for all $i = 1, \ldots, n$. Then, $\boldsymbol{\Lambda}(\cdot)$ admits the (functional) singular value decomposition

$$\boldsymbol{\Lambda}(\cdot) = \sum_{i=1}^n \sigma_i u_i v_i(\cdot).$$

Define $\bar{\mathcal{A}}$ and its corresponding parameters analogously. By definition,

$$\left\|\bar{\mathcal{A}} - \mathcal{A}\right\|_2 \leq \sup_{t \in \mathcal{T}} \left\|\bar{\boldsymbol{\Lambda}}(t) - \boldsymbol{\Lambda}(t)\right\|_2 \leq \sup_{t \in \mathcal{T}} \max_{i,j \in [n]} \left|\bar{\lambda}_{ij}(t) - \lambda_{ij}(t)\right| \leq \frac{nL}{M}.$$

Therefore, by (a functional version of) Wedin's $\sin \Theta$ theorem

$$\left\|\bar{\mathbf{U}}\mathbf{W}_1 - \mathbf{U}\right\|_2 \lesssim \frac{\left\|\bar{\mathcal{A}} - \mathcal{A}\right\|_2}{\sigma_d - \sigma_{d+1}} \leq \frac{nL}{M\delta}.$$

### F.6.3 Controlling the bias term

Combining the above bounds, we have that uniformly for all $i, j, t$,

$$
\begin{aligned}
\left\|\bar{Y}_i(t)\mathbf{W}_1 - Y_i(t)\right\|_2 &= \left\|\bar{\boldsymbol{\Lambda}}(t)\bar{\mathbf{U}}\mathbf{W}_1 - \boldsymbol{\Lambda}(t)\mathbf{U}\right\|_2 \\
&\leq \left\|\bar{\boldsymbol{\Lambda}}(t)\right\|_{2,\infty} \left\|\bar{\mathbf{U}}\mathbf{W}_2 - \mathbf{U}\right\|_2 + \left\|\bar{\boldsymbol{\Lambda}}(t) - \boldsymbol{\Lambda}(t)\right\|_{2,\infty} \left\|\mathbf{U}\right\|_2 \\
&\lesssim \frac{n^{3/2}\lambda_{\max}L}{M\delta} + \frac{\sqrt{n}L}{M} \\
&\leq \frac{n^{3/2}\lambda_{\max}L}{M\delta},
\end{aligned}
$$

where the final inequality follows from the fact that $\delta \leq n\lambda_{\max}$.

## G   Proofs of the technical propositions

### G.1   Proof of Proposition 2

We have that $M^{-1}$ times the lower-triangular elements of each block of $\widehat{\boldsymbol{\Lambda}}$ are independent Poisson random variables with mean given by $M^{-1}$ times the lower-triangular elements of each block of $\bar{\boldsymbol{\Lambda}}$. Define the matrices $\widehat{\boldsymbol{\Lambda}}^{\mathrm{L}}$ and $\widehat{\boldsymbol{\Lambda}}^{\mathrm{U}}$ with the upper and lower triangles, respectively, of each block set to zero, and the diagonals of each block halved, and define $\bar{\boldsymbol{\Lambda}}^{\mathrm{L}}$ and $\bar{\boldsymbol{\Lambda}}^{\mathrm{U}}$ similarly, so that $M^{-1}\widehat{\boldsymbol{\Lambda}}^{\mathrm{L}}$ (respectively $M^{-1}\widehat{\boldsymbol{\Lambda}}^{\mathrm{U}}$) has independent Poisson entries with means $M^{-1}\bar{\boldsymbol{\Lambda}}^{\mathrm{L}}$ (respectively $M^{-1}\bar{\boldsymbol{\Lambda}}^{\mathrm{U}}$), and $\widehat{\boldsymbol{\Lambda}} - \bar{\boldsymbol{\Lambda}} = (\widehat{\boldsymbol{\Lambda}}^{\mathrm{L}} - \bar{\boldsymbol{\Lambda}}^{\mathrm{L}}) + (\widehat{\boldsymbol{\Lambda}}^{\mathrm{U}} - \bar{\boldsymbol{\Lambda}}^{\mathrm{U}})$.

We condition on the event that $(\widehat{\boldsymbol{\Lambda}}^{\mathrm{L}} - \bar{\boldsymbol{\Lambda}}^{\mathrm{L}})_{ij} \lesssim M \log n$ for all $i, j$, which occurs with overwhelming probability by Lemma 3 and a union bound. Now, we employ Lemma 5 with $B := M \log n$ and $\nu := Mn\lambda_{\max}$ to obtain

$$\mathbb{P}\left(\left\|\widehat{\boldsymbol{\Lambda}}^{\mathrm{L}} - \bar{\boldsymbol{\Lambda}}^{\mathrm{L}}\right\|_2 \geq 4\sqrt{Mn\lambda_{\max}} + t\right) \leq n \exp\left(-\frac{t^2}{c(M \log n)^2}\right).$$

Setting $t = M \log^{3/2} n$, we have that

$$\left\|\widehat{\boldsymbol{\Lambda}}^{\mathrm{L}} - \bar{\boldsymbol{\Lambda}}^{\mathrm{L}}\right\|_2 \overset{\mathbb{P}}{\lesssim} \sqrt{Mn\lambda_{\max}} + M \log^{3/2} n \lesssim \sqrt{Mn\lambda_{\max}}$$

where the final inequality follows from Assumption 4. We obtain an analogous bound for $\left\|\widehat{\boldsymbol{\Lambda}}^{\mathrm{U}} - \bar{\boldsymbol{\Lambda}}^{\mathrm{U}}\right\|_2$ and combine the with the triangle inequality:

$$\left\|\widehat{\boldsymbol{\Lambda}} - \bar{\boldsymbol{\Lambda}}\right\|_2 \leq \left\|\widehat{\boldsymbol{\Lambda}}^{\mathrm{L}} - \bar{\boldsymbol{\Lambda}}^{\mathrm{L}}\right\|_2 + \left\|\widehat{\boldsymbol{\Lambda}}^{\mathrm{U}} - \bar{\boldsymbol{\Lambda}}^{\mathrm{U}}\right\|_2 \overset{\mathbb{P}}{\lesssim} \sqrt{Mn\lambda_{\max}}.$$

We now establish a bound on $\left\|\widehat{\boldsymbol{\Lambda}} - \bar{\boldsymbol{\Lambda}}\right\|_1$. We condition on the event $|\widehat{\boldsymbol{\Lambda}}_{ij} - \bar{\boldsymbol{\Lambda}}_{ij}| \lesssim M \log n$ for all $i, j$, which occurs with overwhelming probability due to Lemma 3 and a union bound, and note that we have $\sum_{j=1}^n \mathbb{E}(\widehat{\boldsymbol{\Lambda}}_{ji} - \bar{\boldsymbol{\Lambda}}_{ji})^2 \leq Mn\lambda_{\max}$. Then, by the classical Bernstein inequality, we have for any $t > 0$,

$$\mathbb{P}\left\{\sum_{j=1}^n \left|\widehat{\boldsymbol{\Lambda}}_{ji} - \bar{\boldsymbol{\Lambda}}_{ji}\right| \geq t\right\} \leq 2 \exp\left\{\frac{-t^2}{2\left(Mn\lambda_{\max} + tM \log n/3\right)}\right\},$$

and setting $t = \sqrt{nM\lambda_{\max}\log n}$, we obtain

$$\sum_{j=1}^{n}\left|\widehat{\boldsymbol{\Lambda}}_{ji} - \bar{\boldsymbol{\Lambda}}_{ji}\right| \overset{\mathbb{P}}{\lesssim} \sqrt{nM\lambda_{\max}\log n}.$$

A union bound establishes that

$$\left\|\widehat{\boldsymbol{\Lambda}} - \bar{\boldsymbol{\Lambda}}\right\|_{1} \overset{\mathbb{P}}{\lesssim} \sqrt{nM\lambda_{\max}\log n},$$

which establishes Proposition 2.

## G.2  Proof of Proposition 3

Proposition 3 follows from an application of Weyl's inequality. We have

$$\sigma_1(\widehat{\boldsymbol{\Lambda}}) \leq \sigma_1(\bar{\boldsymbol{\Lambda}}) + |\sigma_1(\widehat{\boldsymbol{\Lambda}}) - \sigma_1(\bar{\boldsymbol{\Lambda}})| \leq \sigma_1(\bar{\boldsymbol{\Lambda}}) + \left\|\widehat{\boldsymbol{\Lambda}} - \bar{\boldsymbol{\Lambda}}\right\|_2 \overset{\mathbb{P}}{\lesssim} \sqrt{M\sigma_1(\boldsymbol{\Sigma})} + \sqrt{Mn\lambda_{\max}} \lesssim \sqrt{M\sigma_1(\boldsymbol{\Sigma})}$$

since $\sigma_1(\bar{\boldsymbol{\Lambda}}) = \sqrt{M\sigma_1(\boldsymbol{\Sigma})} \gtrsim \sqrt{M\delta} \gtrsim \sqrt{Mn\lambda_{\max}}$. Similarly, we have

$$\sigma_d(\widehat{\boldsymbol{\Lambda}}) \geq \sigma_d(\bar{\boldsymbol{\Lambda}}) - |\sigma_1(\widehat{\boldsymbol{\Lambda}}) - \sigma_1(\bar{\boldsymbol{\Lambda}})| \geq \sigma_d(\bar{\boldsymbol{\Lambda}}) - \left\|\widehat{\boldsymbol{\Lambda}} - \bar{\boldsymbol{\Lambda}}\right\|_2 \overset{\mathbb{P}}{\gtrsim} \sqrt{M\sigma_d(\boldsymbol{\Sigma})} - \sqrt{Mn\lambda_{\max}} \gtrsim \sqrt{M\sigma_d(\boldsymbol{\Sigma})},$$

which establishes the proposition.

## G.3  Proof of Proposition 4

We begin by constructing matrices $\bar{\mathbf{Q}}$ and $\bar{\mathbf{E}}$, via a symmetric dilation trick, such that the spectral norms of $\mathbf{Q}^{\top}(\widehat{\boldsymbol{\Lambda}} - \bar{\boldsymbol{\Lambda}})\mathbf{R}$ and $\bar{\mathbf{Q}}^{\top}\bar{\mathbf{E}}\bar{\mathbf{Q}}$ coincide, and then apply a classical $\varepsilon$-net argument to the spectral norm of $\bar{\mathbf{Q}}^{\top}\bar{\mathbf{E}}\bar{\mathbf{Q}}$, following the proof of Lemma D.1 in [66].

First, we set $\bar{\mathbf{E}} := \mathcal{D}(\widehat{\boldsymbol{\Lambda}} - \bar{\boldsymbol{\Lambda}})$, where $\mathcal{D}$ is the dilation operator (see Section F.1.2) and $\bar{\mathbf{Q}} = (\mathbf{Q}\ \mathbf{R})$, and observe that

$$\left\|\mathbf{Q}^{\top}\left(\widehat{\boldsymbol{\Lambda}} - \bar{\boldsymbol{\Lambda}}\right)\mathbf{R}\right\|_2 = \left\|\bar{\mathbf{Q}}^{\top}\bar{\mathbf{E}}\bar{\mathbf{Q}}\right\|_2 = \max_{\|v\|_2 \leq 1}\left|v^{\top}\bar{\mathbf{Q}}^{\top}\bar{\mathbf{E}}\bar{\mathbf{Q}}v\right|$$

where the second equality follows from the Courant-Fischer min-max theorem. Now, let $\mathcal{S}_{\varepsilon}^{d-1}$ be an $\varepsilon$-net of the $d-1$–dimensional unit sphere $\mathcal{S}^{d-1} := \{v : \|v\|_2 = 1\}$. By definition, for any $v \in \mathcal{S}^{d-1}$, there exists some $w(v) \in \mathcal{S}_{\varepsilon}^{d-1}$ such that $\|v - w(v)\|_2 < \varepsilon$ and

$$\begin{aligned}
\left\|\bar{\mathbf{Q}}^{\top}\bar{\mathbf{E}}\bar{\mathbf{Q}}\right\|_2 &= \max_{\|v\|_2 \leq 1}\left|v^{\top}\bar{\mathbf{Q}}^{\top}\bar{\mathbf{E}}\bar{\mathbf{Q}}v\right| \\
&= \max_{\|v\|_2 \leq 1}\left|\left\{v^{\top} - w(v) + w(v)\right\}^{\top}\bar{\mathbf{Q}}^{\top}\bar{\mathbf{E}}\bar{\mathbf{Q}}\left\{v - w(v) + w(v)\right\}\right| \\
&\leq \left(\varepsilon^2 + 2\varepsilon\right)\left\|\bar{\mathbf{Q}}^{\top}\bar{\mathbf{E}}\bar{\mathbf{Q}}\right\|_2 + \max_{w \in \mathcal{S}_{\varepsilon}^{d-1}}\left|w^{\top}\bar{\mathbf{Q}}^{\top}\bar{\mathbf{E}}\bar{\mathbf{Q}}w\right|.
\end{aligned}$$

With $\varepsilon = 1/3$, we have

$$\left\|\bar{\mathbf{Q}}^{\top}\bar{\mathbf{E}}\bar{\mathbf{Q}}\right\|_2 \leq \frac{9}{2}\max_{w \in \mathcal{S}_{\varepsilon}^{d-1}}\left|w^{\top}\bar{\mathbf{Q}}^{\top}\bar{\mathbf{E}}\bar{\mathbf{Q}}w\right|.$$

Now, $\mathcal{S}_{1/3}^{d-1}$ can be selected so that its cardinality can be upper bounded by $|\mathcal{S}_{1/3}^{d-1}| \leq 18^d$ (see, for example, Pollard [70]). For a fixed $w \in \mathcal{S}_{1/3}^{d-1}$, we let $z = \bar{\mathbf{Q}}w$ and note that since $\mathcal{S}_{1/3}^{d-1} \subset \mathcal{S}^{d-1}$, that $\|z\|_2 \leq 1$, and

$$\left|w^{\top}\bar{\mathbf{Q}}^{\top}\bar{\mathbf{E}}\bar{\mathbf{Q}}w\right| = \left|\sum_{i=1}^{n(M+1)}\sum_{j=1}^{n(M+1)}\bar{e}_{ij}z_i z_j\right| = 2\left|\sum_{i=1}^{n}\sum_{j=1}^{nM}e_{ij}z_i z_{n+j}\right|$$

Now, over the event that entries $e_{ij} \lesssim M \log n$, for all $i, j$, which occurs which overwhelming probability by Lemma 3, Hoeffding's inequality and a union bound over $w \in \mathcal{S}_{1/3}^{d-1}$ gives

$$\mathbb{P}\left\{\left\|\bar{\mathbf{Q}}^\top \bar{\mathbf{E}} \bar{\mathbf{Q}}\right\|_2 > t\right\} \leq \sum_{w \in \mathcal{S}_{1/3}^{d-1}} \mathbb{P}\left(\left|w^\top \mathbf{Q}^\top \bar{\mathbf{E}} \bar{\mathbf{Q}} w\right| > \frac{2t}{9}\right)$$

$$= \sum_{w \in \mathcal{S}_{1/3}^{d-1}} \mathbb{P}\left\{\left|\sum_{i=1}^{n}\sum_{j=1}^{nM} e_{ij} z_i z_{n+j}\right| > \frac{t}{9}\right\}$$

$$\leq 2 \cdot 18^d \exp\left\{-\frac{2t^2}{(9cM\log n)^2}\right\}$$

$$= 2 \cdot \exp\left\{d\log(18) - \frac{2t^2}{(9cM\log n)^2}\right\}$$

Setting $t = M \log^{3/2} n$ gives

$$\left\|\mathbf{Q}^\top\left(\widehat{\mathbf{\Lambda}} - \tilde{\mathbf{\Lambda}}\right)\mathbf{R}\right\|_2 = \left\|\bar{\mathbf{Q}}^\top \bar{\mathbf{E}} \bar{\mathbf{Q}}\right\|_2 \overset{\mathbb{P}}{\lesssim} M\log^{3/2} n,$$

completing the proof.

### G.4 Proof of Proposition 5

Denote the singular value decomposition of $\bar{\mathbf{U}}^\top \widehat{\mathbf{U}}$ by $\mathbf{\Omega}_1 \mathbf{\Xi} \mathbf{\Omega}_2^\top$, where $\mathbf{\Xi} = \mathrm{diag}(\xi_1, \ldots, \xi_d)$, and let $\mathbf{W} := \mathbf{\Omega}_1 \mathbf{\Omega}_2^\top$. The principal angles $\{\theta_i\}_{i=1}^d$ between the column spaces of $\bar{\mathbf{U}}$ and $\widehat{\mathbf{U}}$ are defined by $\xi_i = \cos(\theta_i)$, and by the Wedin $\sin\Theta$ theorem, we have

$$\left\|\bar{\mathbf{U}}^\top \widehat{\mathbf{U}} - \mathbf{W}\right\|_2 = \|\mathbf{\Xi} - \mathbf{I}\|_2 = \max_{i \in [d]} |1 - \xi_i| = \max_{i \in [d]} |1 - \cos\theta_i| \leq \max_{i \in [d]} |1 - \cos^2\theta_i|$$

$$= \max_{i \in [d]} \sin^2\theta_i \lesssim \frac{\|\widehat{\mathbf{\Lambda}} - \bar{\mathbf{\Lambda}}\|_2^2}{(\sigma_d(\bar{\mathbf{\Lambda}}) - \sigma_{d+1}(\bar{\mathbf{\Lambda}}))^2} \overset{\mathbb{P}}{\lesssim} \frac{Mn\lambda_{\max}}{M\delta^2} = \frac{n\lambda_{\max}}{\delta^2} \lesssim \frac{\sqrt{n\lambda_{\max}}}{\delta}. \tag{21}$$

We apply the Wedin $\sin\Theta$ theorem again to obtain a bound which we will require later:

$$\left\|\widehat{\mathbf{U}}\widehat{\mathbf{U}}^\top - \bar{\mathbf{U}}\bar{\mathbf{U}}^\top\right\|_2 \vee \left\|\widehat{\mathbf{V}}\widehat{\mathbf{V}}^\top - \bar{\mathbf{V}}\bar{\mathbf{V}}^\top\right\|_2 = \left\|\sin\Theta\left(\widehat{\mathbf{U}}, \bar{\mathbf{U}}\right)\right\|_2 \vee \left\|\sin\Theta\left(\widehat{\mathbf{V}}, \bar{\mathbf{V}}\right)\right\|_2$$

$$\lesssim \frac{\|\widehat{\mathbf{\Lambda}} - \bar{\mathbf{\Lambda}}\|_2}{\sigma_d(\bar{\mathbf{\Lambda}}) - \sigma_{d+1}(\bar{\mathbf{\Lambda}})}$$

$$\overset{\mathbb{P}}{\lesssim} \frac{\sqrt{n\lambda_{\max}}}{\delta}.$$

We now establish a bound on $\left\|\bar{\mathbf{U}}^\top \widehat{\mathbf{U}} - \bar{\mathbf{V}}^\top \widehat{\mathbf{V}}\right\|_2$. We start by showing that

$$\left\|\bar{\mathbf{U}}^\top \widehat{\mathbf{U}} - \bar{\mathbf{V}}^\top \widehat{\mathbf{V}}\right\|_2 = \underset{x:\|x\|_2 \leq 1}{\arg\max} \, x^\top \left(\bar{\mathbf{U}}^\top \widehat{\mathbf{U}} - \bar{\mathbf{V}}^\top \widehat{\mathbf{V}}\right) x$$

$$= \underset{x:\|x\|_2 \leq 1}{\arg\max} \sum_{i,j=1}^{d} x_i x_j \left(\bar{\mathbf{U}}^\top \widehat{\mathbf{U}} - \bar{\mathbf{V}}^\top \widehat{\mathbf{V}}\right)_{ij}$$

$$\leq \underset{x:\|x\|_2 \leq 1}{\arg\max} \sum_{i,j=1}^{d} (1 + \bar{s}_i) x_i (1 + \bar{s}_j^{-1}) x_j \left(\bar{\mathbf{U}}^\top \widehat{\mathbf{U}} - \bar{\mathbf{V}}^\top \widehat{\mathbf{V}}\right)_{ij}$$

$$= \underset{x:\|x\|_2 \leq 1}{\arg\max} \sum_{i,j=1}^{d} x^\top \left[\left(\bar{\mathbf{U}}^\top \widehat{\mathbf{U}} - \bar{\mathbf{V}}^\top \widehat{\mathbf{V}}\right) + \bar{\mathbf{S}}\left(\bar{\mathbf{U}}^\top \widehat{\mathbf{U}} - \bar{\mathbf{V}}^\top \widehat{\mathbf{V}}\right)\widehat{\mathbf{S}}^{-1}\right] x$$

$$= \left\|\left(\bar{\mathbf{U}}^\top \widehat{\mathbf{U}} - \bar{\mathbf{V}}^\top \widehat{\mathbf{V}}\right) + \bar{\mathbf{S}}\left(\bar{\mathbf{U}}^\top \widehat{\mathbf{U}} - \bar{\mathbf{V}}^\top \widehat{\mathbf{V}}\right)\widehat{\mathbf{S}}^{-1}\right\|_2,$$

and then we employ the decomposition

$$
\bar{\mathbf{U}}^\top \widehat{\mathbf{U}} - \bar{\mathbf{V}}^\top \widehat{\mathbf{V}} + \bar{\mathbf{S}} \left( \bar{\mathbf{U}}^\top \widehat{\mathbf{U}} - \bar{\mathbf{V}}^\top \widehat{\mathbf{V}} \right) \widehat{\mathbf{S}}^{-1}
$$

$$
= \left[ \bar{\mathbf{U}}^\top \widehat{\mathbf{U}} \widehat{\mathbf{S}} - \bar{\mathbf{S}} \bar{\mathbf{V}}^\top \widehat{\mathbf{V}} + \bar{\mathbf{S}} \bar{\mathbf{U}}^\top \widehat{\mathbf{U}} - \bar{\mathbf{V}} \widehat{\mathbf{V}} \right] \widehat{\mathbf{S}}^{-1}
$$

$$
= \left[ \bar{\mathbf{U}}^\top \left( \widehat{\boldsymbol{\Lambda}} - \bar{\boldsymbol{\Lambda}} \right) \widehat{\mathbf{V}} + \bar{\mathbf{V}}^\top \left( \widehat{\boldsymbol{\Lambda}} - \bar{\boldsymbol{\Lambda}} \right)^\top \widehat{\mathbf{U}} \right] \widehat{\mathbf{S}}^{-1}
$$

$$
= \bar{\mathbf{U}}^\top \left( \widehat{\boldsymbol{\Lambda}} - \bar{\boldsymbol{\Lambda}} \right) \left( \widehat{\mathbf{V}} - \bar{\mathbf{V}} \bar{\mathbf{V}}^\top \widehat{\mathbf{V}} \right) \widehat{\mathbf{S}}^{-1} + \bar{\mathbf{U}}^\top \left( \widehat{\boldsymbol{\Lambda}} - \bar{\boldsymbol{\Lambda}} \right) \bar{\mathbf{V}} \bar{\mathbf{V}}^\top \widehat{\mathbf{V}} \widehat{\mathbf{S}}^{-1}
$$

$$
+ \bar{\mathbf{V}}^\top \left( \widehat{\boldsymbol{\Lambda}} - \bar{\boldsymbol{\Lambda}} \right)^\top \left( \widehat{\mathbf{U}} - \bar{\mathbf{U}} \bar{\mathbf{U}}^\top \widehat{\mathbf{U}} \right) \widehat{\mathbf{S}}^{-1} + \bar{\mathbf{V}}^\top \left( \widehat{\boldsymbol{\Lambda}} - \bar{\boldsymbol{\Lambda}} \right)^\top \bar{\mathbf{U}} \bar{\mathbf{U}}^\top \widehat{\mathbf{U}} \widehat{\mathbf{S}}^{-1}.
$$

Therefore we have

$$
\left\| \bar{\mathbf{U}}^\top \widehat{\mathbf{U}} - \bar{\mathbf{V}}^\top \widehat{\mathbf{V}} \right\|_2 \leq \left\| \widehat{\boldsymbol{\Lambda}} - \bar{\boldsymbol{\Lambda}} \right\|_2 \left( \left\| \widehat{\mathbf{V}} \widehat{\mathbf{V}}^\top - \bar{\mathbf{V}} \bar{\mathbf{V}} \right\|_2 + \left\| \widehat{\mathbf{U}} \widehat{\mathbf{U}}^\top - \bar{\mathbf{U}} \bar{\mathbf{U}} \right\|_2 \right) \left\| \widehat{\mathbf{S}}^{-1} \right\|_2
$$

$$
\left\| \bar{\mathbf{U}}^\top \left( \widehat{\boldsymbol{\Lambda}} - \bar{\boldsymbol{\Lambda}} \right) \bar{\mathbf{V}} \right\|_2 \left\| \widehat{\mathbf{S}}^{-1} \right\|_2 + \left\| \bar{\mathbf{V}}^\top \left( \widehat{\boldsymbol{\Lambda}} - \bar{\boldsymbol{\Lambda}} \right)^\top \bar{\mathbf{U}} \right\|_2 \left\| \widehat{\mathbf{S}}^{-1} \right\|_2
$$

$$
\overset{\mathbb{P}}{\lesssim} \sqrt{Mn\lambda_{\max}} \cdot \frac{\sqrt{n\lambda_{\max}}}{\delta} \cdot \frac{1}{\sigma_d(\bar{\boldsymbol{\Lambda}})} + \frac{M \log^{3/2} n}{\sigma_d(\bar{\boldsymbol{\Lambda}})}
$$

$$
= \frac{n\lambda_{\max}}{\delta^2} + \frac{\sqrt{M} \log^{3/2} n}{\delta}
$$

$$
\lesssim \frac{\sqrt{n\lambda_{\max}}}{\delta}.
$$

Combining this with (21), we have

$$
\left\| \bar{\mathbf{V}}^\top \widehat{\mathbf{V}} - \mathbf{W} \right\|_2 \leq \left\| \bar{\mathbf{V}}^\top \widehat{\mathbf{V}} - \bar{\mathbf{U}}^\top \widehat{\mathbf{U}} \right\|_2 + \left\| \bar{\mathbf{U}}^\top \widehat{\mathbf{U}} - \mathbf{W} \right\|_2 \overset{\mathbb{P}}{\lesssim} \frac{\sqrt{n\lambda_{\max}}}{\delta}.
$$

## G.5 Proof of Proposition 6

We begin by decomposing $\mathbf{W} \widehat{\mathbf{S}} - \bar{\mathbf{S}} \mathbf{W}$ as

$$
\mathbf{W} \widehat{\mathbf{S}} - \bar{\mathbf{S}} \mathbf{W} = \left( \mathbf{W} - \bar{\mathbf{U}}^\top \widehat{\mathbf{U}} \right) \widehat{\mathbf{S}} + \bar{\mathbf{S}} \left( \mathbf{V}^\top \widehat{\mathbf{V}} - \mathbf{W} \right) + \bar{\mathbf{U}}^\top \widehat{\mathbf{U}} \widehat{\mathbf{S}} - \bar{\mathbf{S}} \bar{\mathbf{V}}^\top \widehat{\mathbf{V}}
$$

$$
= \left( \mathbf{W} - \bar{\mathbf{U}}^\top \widehat{\mathbf{U}} \right) \widehat{\mathbf{S}} + \bar{\mathbf{S}} \left( \mathbf{V}^\top \widehat{\mathbf{V}} - \mathbf{W} \right) + \bar{\mathbf{U}}^\top \left( \widehat{\boldsymbol{\Lambda}} - \bar{\boldsymbol{\Lambda}} \right) \widehat{\mathbf{V}}
$$

$$
= \left( \mathbf{W} - \bar{\mathbf{U}}^\top \widehat{\mathbf{U}} \right) \widehat{\mathbf{S}} + \bar{\mathbf{S}} \left( \mathbf{V}^\top \widehat{\mathbf{V}} - \mathbf{W} \right) + \bar{\mathbf{U}}^\top \left( \widehat{\boldsymbol{\Lambda}} - \bar{\boldsymbol{\Lambda}} \right) \left( \widehat{\mathbf{V}} \widehat{\mathbf{V}}^\top - \bar{\mathbf{V}} \bar{\mathbf{V}}^\top \right) \widehat{\mathbf{V}}
$$

$$
+ \bar{\mathbf{U}}^\top \left( \widehat{\boldsymbol{\Lambda}} - \bar{\boldsymbol{\Lambda}} \right) \bar{\mathbf{V}} \bar{\mathbf{V}}^\top \widehat{\mathbf{V}},
$$

and therefore we have that

$$
\left\| \mathbf{W} \widehat{\mathbf{S}} - \bar{\mathbf{S}} \mathbf{W} \right\|_2 \leq \left\| \mathbf{W} - \bar{\mathbf{U}}^\top \widehat{\mathbf{U}} \right\|_2 \left\| \widehat{\boldsymbol{\Lambda}} \right\|_2 + \left\| \mathbf{W} - \bar{\mathbf{V}}^\top \widehat{\mathbf{V}} \right\|_2 \left\| \bar{\boldsymbol{\Lambda}} \right\|_2
$$

$$
+ \left\| \widehat{\boldsymbol{\Lambda}} - \bar{\boldsymbol{\Lambda}} \right\|_2 \left\| \widehat{\mathbf{V}} \widehat{\mathbf{V}}^\top - \bar{\mathbf{V}} \bar{\mathbf{V}}^\top \right\|_2 + \left\| \bar{\mathbf{U}}^\top \left( \widehat{\boldsymbol{\Lambda}} - \bar{\boldsymbol{\Lambda}} \right) \bar{\mathbf{V}} \right\|_2
$$

$$
\overset{\mathbb{P}}{\lesssim} \frac{\sqrt{n\lambda_{\max}} \kappa}{\delta} + \frac{\sqrt{M} n\lambda_{\max}}{\delta} + M \log^{3/2} n
$$

$$
\lesssim M \log^{3/2} n,
$$

which completes the proof.

## G.6 Proof of Proposition 7

A key tool in proving Proposition 7 is a theorem due [11], providing entrywise eigenvector bounds for random matrices. The original statement is given for the eigenvectors of symmetric random matrices

with row and column-wise independence. We state a generalisation for the singular vectors of rectangular matrices with *block-wise* independence structure. The extension to block-wise independence structure has been handled in [12] (see Proposition 2.1(b) of that paper), although the exposition of the results in this paper is more complicated. For this reason, we choose to state the result due to [11] with this generalisation, which can be seen by following through the relevant parts of their proof.

**Lemma 8** (A slight generalisation of Theorem 2.1 of [11]). *Let $\mathbf{M}_0$ be an $n_1 \times n_2$ real-valued random matrix. Define $n_0 = n_1 + n_2$ and let $\pi_1 = \sqrt{2n_1/n_0}$ and $\pi_2 = \sqrt{2n_2/n_0}$. Define $\kappa_0 := \sigma_1(\mathbb{E}\mathbf{M}_0)/\sigma_d(\mathbb{E}\mathbf{M}_0)$, $\delta_0 = \sigma_d(\mathbb{E}\mathbf{M}_0) - \sigma_{d+1}(\mathbb{E}\mathbf{M}_0)$. Suppose there exists some $\gamma > 0$ and a function $\varphi : \mathbb{R}_+ \to \mathbb{R}_+$ which is continuous and non-decreasing on $\mathbb{R}_+$, with $\varphi(0) = 0$ and $\varphi(x)/x$ non-increasing on $\mathbb{R}_+$, such that the following conditions hold:*

**B1** (Incoherence). $\|\mathbb{E}\mathbf{M}_0\|_{2,\infty} \vee \|\mathbb{E}\mathbf{M}_0^\top\|_{2,\infty} \leq \gamma\delta_0$.

**B2** (Block-wise independence). Assume that for any $k \in [n_1], \ell \in [n_2]$, there exists $\mathcal{N}_k^1 \subset [n_1]$ and $\mathcal{N}_\ell^2 \subset [n_2]$, such that the $k$th row of $\mathbf{M}_0$ is independent of the columns $\{j : j \notin \mathcal{N}_k^1\}$, and the $\ell$th column of $\mathbf{M}_0$ is independent of the rows $\{i : i \notin \mathcal{N}_\ell^2\}$. Let $m_0 = \max_{k,\ell}\left\{|\mathcal{N}_k^1| \vee |\mathcal{N}_\ell^2|\right\}$ and assume $m_0 \lesssim \delta_0$.

**B3** (Spectral norm concentration). $\kappa_0 \max\{\gamma, \varphi(\gamma)\} \lesssim 1$ and $\mathbb{P}(\|\mathbf{M}_0 - \mathbb{E}\mathbf{M}_0\|_2 > \gamma\Delta) \leq \eta_0$ for some $\eta_0 \in (0,1)$.

**B4.** [Row and column concentration] There exists some $\eta_1 \in (0,1)$ such that for any matrices $\mathbf{Q} \in \mathbb{R}^{n_1 \times d}, \mathbf{R} \in \mathbb{R}^{n_2 \times d}$ and $i \in [n_1], j \in [n_2]$,

$$\mathbb{P}\left\{\left\|(\mathbf{M}_0 - \mathbb{E}\mathbf{M}_0)_{\cdot,i}\,\mathbf{R}\right\|_2 \leq \delta_0 b_\infty \varphi\left(\frac{b_{\mathrm{F}}}{\sqrt{n_0}b_\infty}\right)\right\} \geq 1 - \frac{\eta_1}{n_0},$$

and

$$\mathbb{P}\left\{\left\|(\mathbf{M}_0 - \mathbb{E}\mathbf{M}_0)_{j,\cdot}\,\mathbf{Q}\right\|_2 \leq \delta_0 b_\infty \varphi\left(\frac{b_{\mathrm{F}}}{\sqrt{n_0}b_\infty}\right)\right\} \geq 1 - \frac{\eta_1}{n_0}$$

where $b_\infty := \pi_1\|\mathbf{Q}\|_{2,\infty} \vee \pi_2\|\mathbf{R}\|_{2,\infty}$, and $b_{\mathrm{F}} := \left(\pi_1\|\mathbf{Q}\|_{\mathrm{F}}^2 + \pi_2\|\mathbf{R}\|_{\mathrm{F}}^2\right)^{1/2}$.

*Let $\widehat{\mathbf{U}}_0, \mathbf{U}_0$ (respectively $\widehat{\mathbf{V}}_0, \mathbf{V}_0$) be the matrices containing the left (respectively, right) singular vectors corresponding to the $d$ leading singular values of $\mathbf{M}_0$ and $\mathbb{E}\mathbf{M}_0$. Then, with probability at least $1 - \eta_0 - 2\eta_1$, we have*

$$\pi_1\|\widehat{\mathbf{U}}_0\|_{2,\infty} \vee \pi_2\|\widehat{\mathbf{V}}_0\|_{2,\infty} \lesssim \{\kappa_0 + \varphi(1)\}\,(\pi_1\|\mathbf{U}_0\|_{2,\infty} \vee \pi_2\|\mathbf{V}_0\|_{2,\infty})$$
$$+ \gamma(\pi_1\|\mathbb{E}\mathbf{M}_0\|_{2,\infty} \vee \pi_2\|(\mathbb{E}\mathbf{M}_0)^\top\|_{2,\infty})/\delta_0;$$

$$\pi_1\|\widehat{\mathbf{U}}_0\mathbf{O} - \mathbf{U}_0\|_{2,\infty} \vee \pi_2\|\widehat{\mathbf{V}}_0\mathbf{O} - \mathbf{V}_0\|_{2,\infty} \lesssim [\kappa_0\{\kappa_0 + \varphi(1)\}\{\gamma + \varphi(\gamma)\} + \varphi(1)]\,(\pi_1\|\mathbf{U}_0\|_{2,\infty} \vee \pi_2\|\mathbf{V}_0\|_{2,\infty})$$
$$+ \gamma(\pi_1\|\mathbb{E}\mathbf{M}_0\|_{2,\infty} \vee \pi_2\|(\mathbb{E}\mathbf{M}_0)^\top\|_{2,\infty})/\delta_0.$$

The following is an adaptation of Lemma D.2 of [66] (see also Lemma 7 of [11]) who showed an analogous result for Bernoulli random variables.

**Lemma 9.** *Let $Y_i \sim \mathrm{Poisson}(\lambda_i)$ independently for all $i = 1,\ldots,n$, and suppose $\mathbf{Q}$ is a deterministic matrix. The $Q_i$ denote the $i$th row of $\mathbf{Q}$, and set $\lambda_{\max} := \max_{i\in[n]}\lambda_i$. Then for any $\alpha > 0$,*

$$\mathbb{P}\left\{\left\|\sum_{i=1}^n (Y_i - \lambda_i)\,Q_i\right\| > \frac{(2+\alpha)n\lambda_{\max}\|\mathbf{Q}\|_{2,\infty}}{1 \vee \log\left(\sqrt{n}\,\|\mathbf{Q}\|_{2,\infty}/\|\mathbf{Q}\|_{\mathrm{F}}\right)}\right\} \leq 2de^{-\alpha n\lambda_{\max}}.$$

We omit the proof of Lemma 9, which is identical to the proof of Lemma D.2 of [66] with the Bernoulli moment generating function with the Poisson moment generating function.

With these tools to hand, we begin by obtaining a bound on $\|\widehat{\mathbf{U}}\|_{2,\infty}$ using Lemma 8, with $\mathbf{M}_0 := \widehat{\mathbf{\Lambda}}$. We set $\gamma := \sqrt{n\lambda_{\max}}/\delta$ and

$$\varphi(x) := \frac{n\lambda_{\max}}{\delta\{1 \vee \log(1/x)\}}.$$

First observe that $n_0 = n + nM \asymp nM$ and $\pi_1 \asymp M^{-1/2}$ and $\pi_2 \asymp 1$, and that $\kappa_0 = \kappa$ and $\delta_0 = \sqrt{M}\delta$. **B1** holds since $\|\bar{\boldsymbol{\Lambda}}\|_{2,\infty} \leq \sqrt{n}\lambda_{\max} \lesssim \sqrt{n\lambda_{\max}}$ since $\lambda_{\max} \lesssim 1$ by Assumption 1. Using Assumptions 3 and 4, we have

$$M \lesssim \frac{n\lambda_{\max}}{\log^3 n} \lesssim \frac{\delta \log(\delta/\sqrt{n\lambda_{\max}})}{\kappa \log^3 n} \lesssim \frac{\delta \log n}{\kappa \log^3 n} \lesssim \delta,$$

and therefore **B2** holds. **B3** holds from Proposition 2, and observing that by Assumption 3, $\kappa_0 \max\{\gamma, \phi(\gamma)\} \lesssim 1$.

To see that **B4** holds, note that each row and column of $M^{-1}(\widehat{\boldsymbol{\Lambda}} - \bar{\boldsymbol{\Lambda}})$ contains independent Poisson random variables with means not exceeding $n\lambda_{\max}/M$. Then for $\mathbf{Q} \in \mathbb{R}^{n \times d}$, $\mathbf{R} \in \mathbb{R}^{nM \times d}$, setting $\alpha = \log n/n\lambda_{\max}$ in Lemma 9 implies that

$$\left\| \left(\widehat{\boldsymbol{\Lambda}} - \bar{\boldsymbol{\Lambda}}\right)_{i,\cdot} \mathbf{R} \right\|_{2,\infty} = M \left\| \frac{1}{M} \left(\widehat{\boldsymbol{\Lambda}} - \bar{\boldsymbol{\Lambda}}\right)_{i,\cdot} \mathbf{R} \right\|_{2,\infty}$$

$$\overset{\mathbb{P}}{\lesssim} M \cdot \frac{(n\lambda_{\max}/M + \log n)\, \|\mathbf{R}\|_{2,\infty}}{1 \vee \log\left(\frac{\sqrt{n_0}\|\mathbf{R}\|_{2,\infty}}{\|\mathbf{R}\|_F}\right)}$$

$$\lesssim \frac{n\lambda_{\max} \|\mathbf{R}\|_{2,\infty}}{1 \vee \log\left(\frac{\sqrt{n_0}\|\mathbf{R}\|_{2,\infty}}{\|\mathbf{R}\|_F}\right)}$$

$$= \delta \|\mathbf{R}\|_{2,\infty}\, \varphi\left(\frac{\|\mathbf{R}\|_F}{\sqrt{n_0}\|\mathbf{R}\|_{2,\infty}}\right)$$

$$\leq \delta_0 b_\infty \varphi\left(\frac{b_F}{\sqrt{n_0}b_\infty}\right).$$

Similarly, setting $\alpha = M \log n/n\lambda_{\max}$ we have

$$\left\| \left(\tilde{\mathbf{A}} - \tilde{\boldsymbol{\Lambda}}\right)_{\cdot,i}^\top \mathbf{Q} \right\|_{2,\infty} \overset{\mathbb{P}}{\lesssim} \frac{\sqrt{M} n\lambda_{\max} \|\mathbf{Q}\|_{2,\infty}}{1 \vee \log\left(\sqrt{n_0} \|\mathbf{Q}\|_{2,\infty} / \|\mathbf{Q}\|_F\right)} \leq \delta_0 b_\infty \varphi\left(\frac{\sqrt{n}b_\infty}{b_F}\right),$$

which establishes **B4**. Having established **B1**-**B4**, we are ready to apply Lemma 8:

$$\left\|\widehat{\mathbf{U}}\right\|_{2,\infty} \overset{\mathbb{P}}{\lesssim} \sqrt{M}\{\kappa_0 + \varphi(1)\}(\pi_1\|\mathbf{U}\|_{2,\infty} \vee \pi_2\|\mathbf{V}\|_{2,\infty}) + \sqrt{M}\gamma\left(\pi_1 \|\bar{\boldsymbol{\Lambda}}\|_{2,\infty} \vee \pi_2 \|\bar{\boldsymbol{\Lambda}}^\top\|_{2,\infty}\right)/\delta_0$$

We have

$$\|\bar{\mathbf{V}}\|_{2,\infty} = \|\bar{\boldsymbol{\Lambda}}\bar{\mathbf{U}}\bar{\mathbf{S}}^{-1}\|_{2,\infty} \leq \|\bar{\boldsymbol{\Lambda}}\|_\infty \|\bar{\mathbf{U}}\|_{2,\infty} \|\bar{\mathbf{S}}\|_2^{-1} \lesssim \frac{n\lambda_{\max}\mu\sqrt{d/n}}{\sigma_d^{1/2}(\boldsymbol{\Sigma})}$$

$$\lesssim \frac{n\lambda_{\max}\mu\sqrt{d/n}}{\sqrt{M}\delta} \lesssim \sqrt{\frac{d}{nM}}\mu\log n. \tag{22}$$

where we used Assumption 3 in the final inequality. Therefore

$$\pi_1 \|\bar{\mathbf{U}}\|_{2,\infty} \vee \pi_2 \|\bar{\mathbf{V}}\|_{2,\infty} \leq \sqrt{\frac{d}{nM}}\mu\log n,$$

and we have

$$\kappa = \frac{\sigma_1^{1/2}(\boldsymbol{\Sigma})}{\sigma_d^{1/2}(\boldsymbol{\Sigma})} \lesssim \frac{n\lambda_{\max}}{\delta} = \varphi(1)$$

and so the first term satisfies

$$\sqrt{M}\{\kappa_0 + \varphi(1)\}(\pi_1\|\mathbf{U}\|_{2,\infty} \vee \pi_2\|\mathbf{V}\|_{2,\infty}) \lesssim \sqrt{M}\varphi(1)(\pi_1\|\mathbf{U}\|_{2,\infty} \vee \pi_2\|\mathbf{V}\|_{2,\infty})$$

$$\lesssim \sqrt{M} \cdot \frac{n\lambda_{\max}}{\delta} \cdot \sqrt{\frac{d}{nM}}\mu\log n$$

$$= \frac{\mu\lambda_{\max}\sqrt{nd}\log n}{\delta}.$$

To control the second term, we first observe that

$$\pi_1 \left\| \bar{\boldsymbol{\Lambda}} \right\|_{2,\infty} \leq M^{-1/2} \left\| \bar{\mathbf{U}} \bar{\mathbf{U}}^\top \bar{\boldsymbol{\Lambda}} \right\|_{2,\infty} + M^{-1/2} \left\| \left( \mathbf{I} - \bar{\mathbf{U}} \bar{\mathbf{U}} \right) \bar{\boldsymbol{\Lambda}} \right\|_{2,\infty}$$

$$\lesssim M^{-1/2} \left\| \bar{\mathbf{U}} \right\|_{2,\infty} \left\| \bar{\boldsymbol{\Lambda}} \right\|_2 + M^{-1/2} \max_{i \in [n]} \sup_{t \in (0,1]} r_i(t)$$

$$\lesssim \sqrt{\frac{d}{Mn}} \mu \cdot \sqrt{M} \kappa \delta + \mu \sqrt{d \lambda_{\max}} \log^{5/2} n$$

$$\lesssim \mu \delta \sqrt{d \lambda_{\max}}$$

where the final inequality follows from $\kappa \lesssim \log n \leq \sqrt{n}$ and $\lambda_{\max} \lesssim 1$. Similarly we obtain $\pi_2 \| \bar{\boldsymbol{\Lambda}}^\top \|_{2,\infty} \lesssim \mu \delta \sqrt{d \lambda_{\max}} \log n$ using (22), and therefore

$$\pi_1 \left\| \bar{\boldsymbol{\Lambda}} \right\|_{2,\infty} \vee \pi_2 \left\| \bar{\boldsymbol{\Lambda}}^\top \right\|_{2,\infty} \lesssim \mu \delta \sqrt{d \lambda_{\max}} \log n.$$

We then have

$$\sqrt{M} \gamma \left( \pi_1 \left\| \bar{\boldsymbol{\Lambda}} \right\|_{2,\infty} \vee \pi_2 \left\| \bar{\boldsymbol{\Lambda}}^\top \right\|_{2,\infty} \right) / \delta_0 \lesssim \sqrt{M} \cdot \frac{\sqrt{n \lambda_{max}}}{\delta} \cdot \mu \delta \sqrt{d \lambda_{\max}} \log n \cdot \frac{1}{\sqrt{M} \delta}$$

$$\lesssim \frac{\mu \sqrt{nd} \lambda_{\max} \log n}{\delta}.$$

Combining these bounds, we obtain

$$\left\| \widehat{\mathbf{U}} \right\|_{2,\infty} \overset{\mathbb{P}}{\lesssim} \frac{\mu \sqrt{nd} \lambda_{\max} \log n}{\delta}.$$

We now apply Lemma 8 with $\mathbf{M}_0 = \widehat{\boldsymbol{\Lambda}}^{(m)}$. We set $\gamma$ and $\varphi(x)$ as before and verify Assumptions **B1**-**B4** in the same way. By analogous calculations to the above, we obtain the bound

$$\left\| \widehat{\mathbf{U}}^{(m)} \right\|_{2,\infty} \overset{\mathbb{P}}{\lesssim} \frac{\mu \sqrt{nd} \lambda_{\max} \log n}{\delta}.$$

The final bound is shown in the same way, requiring the additional observation that $\kappa \{ \gamma \vee \varphi(\gamma) \} \lesssim 1$ which follows from Assumption 3, and we obtain

$$\pi_1 \| \widehat{\mathbf{U}}_0 \mathbf{O} - \mathbf{U}_0 \|_{2,\infty} \vee \pi_2 \| \widehat{\mathbf{V}}_0 \mathbf{O} - \mathbf{V}_0 \|_{2,\infty}$$

$$\overset{\mathbb{P}}{\lesssim} \left[ \kappa_0 \{ \kappa_0 + \varphi(1) \} \{ \gamma + \varphi(\gamma) \} + \varphi(1) \right] (\pi_1 \| \mathbf{U}_0 \|_{2,\infty} \vee \pi_2 \| \mathbf{V}_0 \|_{2,\infty}) + \gamma \left( \pi_1 \left\| \bar{\boldsymbol{\Lambda}} \right\|_{2,\infty} \vee \pi_2 \left\| \bar{\boldsymbol{\Lambda}}^\top \right\|_{2,\infty} \right) / \delta_0$$

$$\lesssim \varphi(1) \left( \pi_1 \| \mathbf{U}_0 \|_{2,\infty} \vee \pi_2 \| \mathbf{V}_0 \|_{2,\infty} \right) + \gamma \left( \pi_1 \left\| \bar{\boldsymbol{\Lambda}} \right\|_{2,\infty} \vee \pi_2 \left\| \bar{\boldsymbol{\Lambda}}^\top \right\|_{2,\infty} \right) / \delta_0$$

$$\lesssim \frac{\mu \sqrt{nd} \lambda_{\max} \log n}{\delta}$$

### G.7 Proof of Proposition 8

We show (15) using a simple application of Wedin's inequality:

$$\left\| \widehat{\mathbf{U}}^{(m)} (\widehat{\mathbf{U}}^{(m)})^\top - \bar{\mathbf{U}} \bar{\mathbf{U}}^\top \right\|_2 = \left\| \sin \Theta \left( \widehat{\mathbf{U}}^{(m)}, \bar{\mathbf{U}} \right) \right\|_2 \lesssim \frac{\left\| \widehat{\boldsymbol{\Lambda}}^{(m)} - \bar{\boldsymbol{\Lambda}} \right\|_2}{\sigma_d \left( \bar{\boldsymbol{\Lambda}} \right) - \sigma_{d+1} \left( \bar{\boldsymbol{\Lambda}} \right)} \leq \frac{\left\| \widehat{\boldsymbol{\Lambda}} - \bar{\boldsymbol{\Lambda}} \right\|_2}{\sigma_d \left( \bar{\boldsymbol{\Lambda}} \right) - \sigma_{d+1} \left( \bar{\boldsymbol{\Lambda}} \right)}$$

$$\overset{\mathbb{P}}{\lesssim} \frac{\sqrt{Mn \lambda_{\max}}}{\sqrt{M} \delta} = \frac{\sqrt{n \lambda_{\max}}}{\delta}.$$

The proof of (16) requires a more delicate argument. We apply Wedin's theorem to obtain

$$\left\| \widehat{\mathbf{U}}^{(m)} (\widehat{\mathbf{U}}^{(m)})^\top - \widehat{\mathbf{U}} \widehat{\mathbf{U}}^\top \right\|_2 = \left\| \sin \Theta \left( \widehat{\mathbf{U}}^{(m)}, \widehat{\mathbf{U}} \right) \right\|_2 \lesssim \frac{\left\| \left( \widehat{\boldsymbol{\Lambda}}^{(m)} - \widehat{\boldsymbol{\Lambda}} \right)^\top \widehat{\mathbf{U}}^{(m)} \right\|_2 \vee \left\| \left( \widehat{\boldsymbol{\Lambda}}^{(m)} - \widehat{\boldsymbol{\Lambda}} \right) \widehat{\mathbf{V}}^{(m)} \right\|_2}{\sigma_d \left( \widehat{\boldsymbol{\Lambda}} \right) - \sigma_{d+1} \left( \widehat{\boldsymbol{\Lambda}} \right)}.$$

$$\tag{23}$$

By Weyl's inequality

$$\sigma_d\left(\widehat{\mathbf{\Lambda}}\right) \geq \sigma_d\left(\bar{\mathbf{\Lambda}}\right) + \left\|\widehat{\mathbf{\Lambda}} - \bar{\mathbf{\Lambda}}\right\|_2 \overset{\mathbb{P}}{\gtrsim} \sigma_d\left(\bar{\mathbf{\Lambda}}\right).$$

and

$$\sigma_{d+1}\left(\widehat{\mathbf{\Lambda}}\right) \leq \sigma_{d+1}\left(\bar{\mathbf{\Lambda}}\right) - \left\|\widehat{\mathbf{\Lambda}} - \bar{\mathbf{\Lambda}}\right\|_2 \overset{\mathbb{P}}{\lesssim} \sigma_{d+1}\left(\bar{\mathbf{\Lambda}}\right).$$

and therefore

$$\sigma_d\left(\widehat{\mathbf{\Lambda}}\right) - \sigma_{d+1}\left(\widehat{\mathbf{\Lambda}}\right) \overset{\mathbb{P}}{\gtrsim} \sigma_d\left(\bar{\mathbf{\Lambda}}\right) - \sigma_{d+1}\left(\bar{\mathbf{\Lambda}}\right) = \sqrt{M}\left(\sigma_d^{1/2}(\mathbf{\Sigma}) - \sigma_{d+1}^{1/2}(\mathbf{\Sigma})\right) = \sqrt{M}\delta. \qquad (24)$$

We now focus our attention on obtaining a bound for $\|(\widehat{\mathbf{\Lambda}}^{(m)} - \widehat{\mathbf{\Lambda}})\widehat{\mathbf{U}}^{(m)}\|_{\mathrm{F}}$. Let

$$\mathcal{N}_m = \{m + (\ell - 1)n, \ell \in [M]\}.$$

The $ij$th entry of $\widehat{\mathbf{\Lambda}} - \widehat{\mathbf{\Lambda}}^{(m)}$ is

$$\left(\widehat{\mathbf{\Lambda}}^{(m)} - \widehat{\mathbf{\Lambda}}\right)_{ij} = \left(\widehat{\mathbf{\Lambda}}^{(m)} - \bar{\mathbf{\Lambda}}\right)_{ij} \mathbb{I}\left(i = m, j \in \mathcal{N}_m\right),$$

and so $\widehat{\mathbf{\Lambda}}^{(m)} - \widehat{\mathbf{\Lambda}}$ is independent of $\widehat{\mathbf{\Lambda}}^{(m)}$ and hence $\widehat{\mathbf{\Lambda}}^{(m)} - \widehat{\mathbf{\Lambda}}$ is independent of $\widehat{\mathbf{U}}^{(m)}$. We can then write

$$\left\|\left(\widehat{\mathbf{\Lambda}}^{(m)} - \widehat{\mathbf{\Lambda}}\right)^\top \widehat{\mathbf{U}}^{(m)}\right\|_{\mathrm{F}}^2 = \sum_{\ell \notin \mathcal{N}_m} \left(\widehat{\mathbf{\Lambda}}_{m,\ell} - \bar{\mathbf{\Lambda}}_{m,\ell}\right)^2 \left\|\widehat{\mathbf{U}}_{\ell,\cdot}^{(m)}\right\|_2^2$$

$$+ \left\|\sum_{\ell \in \mathcal{N}_m} \sum_{i=1}^{n} \left(\widehat{\mathbf{\Lambda}}_{i\ell} - \bar{\mathbf{\Lambda}}_{i\ell}\right) \widehat{\mathbf{U}}_{\ell,\cdot}^{(m)}\right\|_2^2$$

$$=: \zeta_1 + \zeta_2.$$

The (square root of the) first term is easily bounded as

$$\zeta_1^{1/2} \leq \left\|\widehat{\mathbf{\Lambda}} - \bar{\mathbf{\Lambda}}\right\|_{2,\infty} \left\|\widehat{\mathbf{U}}^{(m)}\right\|_{2,\infty}$$

$$\leq \left\|\widehat{\mathbf{\Lambda}} - \bar{\mathbf{\Lambda}}\right\|_2 \left\|\widehat{\mathbf{U}}^{(m)}\right\|_{2,\infty}$$

$$\overset{\mathbb{P}}{\lesssim} \sqrt{Mn\lambda_{\max}} \cdot \frac{\mu\sqrt{nd}\lambda_{\max}\log n}{\delta}$$

$$= \frac{\sqrt{M}dn\lambda_{\max}^{3/2}\mu\log n}{\delta}$$

and to bound the second term, we employ Lemma 4 to obtain

$$\zeta_2^{1/2} \overset{\mathbb{P}}{\lesssim} M\log^2 n \left\|\widehat{\mathbf{U}}^{(m)}\right\|_{2,\infty} + \sqrt{M\lambda_{\max}\log n} \left\|\widehat{\mathbf{U}}^{(m)}\right\|_{\mathrm{F}}$$

$$\leq M\log^2 n \left\|\widehat{\mathbf{U}}^{(m)}\right\|_{2,\infty} + \sqrt{M\lambda_{\max}n\log n} \left\|\widehat{\mathbf{U}}^{(m)}\right\|_{2,\infty}$$

$$\lesssim \sqrt{M\lambda_{\max}n\log n} \left\|\widehat{\mathbf{U}}^{(m)}\right\|_{2,\infty}$$

$$\lesssim \sqrt{M\lambda_{\max}n\log n} \cdot \frac{\mu\sqrt{nd}\lambda_{\max}\log n}{\delta}$$

$$= \frac{\sqrt{M}dn\lambda_{\max}^{3/2}\mu\log^{3/2} n}{\delta}$$

where we used Assumption 4 in the third inequality. Therefore

$$\left\|\left(\widehat{\mathbf{\Lambda}}^{(m)} - \widehat{\mathbf{\Lambda}}\right)^\top \widehat{\mathbf{U}}^{(m)}\right\|_2 \leq \left\|\left(\widehat{\mathbf{\Lambda}}^{(m)} - \widehat{\mathbf{\Lambda}}\right)^\top \widehat{\mathbf{U}}^{(m)}\right\|_{\mathrm{F}} \leq \zeta_1^{1/2} + \zeta_2^{1/2} \overset{\mathbb{P}}{\lesssim} \frac{\sqrt{M}dn\lambda_{\max}^{3/2}\mu\log^{3/2} n}{\delta}.$$

Similar analysis yields an analogous bound for $\|(\widehat{\mathbf{\Lambda}}^{(m)} - \widehat{\mathbf{\Lambda}})\widehat{\mathbf{V}}^{(m)}\|_2$, and combining this, with (23) and (24) we have

$$
\left\| \widehat{\mathbf{U}}^{(m)}(\widehat{\mathbf{U}}^{(m)})^{\top} - \widehat{\mathbf{U}}\widehat{\mathbf{U}}^{\top} \right\|_2 \lesssim \frac{\left\| \left(\widehat{\mathbf{\Lambda}}^{(m)} - \widehat{\mathbf{\Lambda}}\right)^{\top} \widehat{\mathbf{U}}^{(m)} \right\|_2 \vee \left\| \left(\widehat{\mathbf{\Lambda}}^{(m)} - \widehat{\mathbf{\Lambda}}\right) \widehat{\mathbf{V}}^{(m)} \right\|_2}{\sigma_d\left(\widehat{\mathbf{\Lambda}}\right) - \sigma_{d+1}\left(\widehat{\mathbf{\Lambda}}\right)}
$$

$$
\overset{\mathbb{P}}{\lesssim} \frac{\sqrt{d}n\lambda_{\max}^{3/2}\mu \log^{3/2} n}{\delta^2}
$$

which establishes the proposition.