# OpenReview forum: "Intensity Profile Projection: A Framework for Continuous-Time Representation Learning for Dynamic Networks"
_NeurIPS.cc/2023/Conference — NeurIPS 2023 poster_

### Official Review · Reviewer_fL1b · 2023-07-03

**Soundness:** 2 fair
**Presentation:** 2 fair
**Contribution:** 2 fair
**Rating:** 3
**Confidence:** 3

**Summary:**

The paper proposes a method to learn low-dimensional continuous-time representations of network nodes, based on the collection of interaction events among them. More precisely, the events are in the form of $(i,j,t)$, where $(i,j)$ is the pair of nodes involved in the interaction event, and $t$ is the occurrence time. The proposed method first estimate the intensity function $\lambda_{i,j}(t)$ of events between each pair of nodes $(i,j)$ at every time instant $t$, then project the intensities of each node at time $t$ onto a learned lower dimensional subspace to obtain a representation. Theoretical results on the recovery error of the representation is provided. Numerical experiments using real data shows the effectiveness of the proposed method.

**Strengths:**

The paper proposes to estimate the representation of nodes using continuous-time events, which seems to be a novel type of data.

**Weaknesses:**

I find the presentation of the paper generally vague and hand-wavy. See the following.

1. The introduction is way too high-level. The authors should be more specific about the problem setting in this paper, for example, why we care about dynamic models, continuous-time event data, low-dimensional representation of nodes etc.

2. The related work is not specific. The authors should use a sentence to summarize the contribution of the mentioned papers and explain the difference from your work.

3. Lemma 1 is not correct. $\widehat U_d$ minimizes the residual sum of squares at $B$ chosen time instants, but not the integrated one.

4. In Section 3, notation part, what is the difference between $\gg$ and $\gtrsim$? Also is the universal constant multiplicative or additive?

5. It's not clear what `$\approx$' means in Section 4.

**Questions:**

.

**Limitations:**

.

---

> ### Author Rebuttal · Authors · 2023-08-05
>
>
> > The introduction is way too high-level. The authors should be more specific about the problem setting in this paper, for example, why we care about dynamic models, continuous-time event data, low-dimensional representation of nodes etc.
>
> See global response.
>
> > The related work is not specific. The authors should use a sentence to summarize the contribution of the mentioned papers and explain the difference from your work.
>
> See global response.
>
> > Lemma 1 is not correct. $\hat U_d$ minimizes the residual sum of squares at $B$ chosen time instants, but not the integrated one.
>
> Lemma 1 is correct as stated. Please see the proof in Section D of the supplementary materials.
>
> > In Section 3, notation part, what is the difference between ≫ and ≳? Also is the universal constant multiplicative or additive?
> Thanks for pointing this out, we will make this explicit in the revision. The symbols $\gg$ and $\gtrsim$ denote the inequalities $>$ and $\geq$ that hide multiplicative constants.
>
> > It's not clear what '≈' means in Section 4.
>
> The "$\approx$ " symbol means "approximately equal to" and is deliberately left informal in the descriptions of the "Structure preserving" and "Temporally coherent" properties. In the paragraph before Lemma 2, $\hat X_i(s) \approx \hat X_j(t)$ is formally defined to mean that $X_i(s) = X_j(t)$, so that the lemma is a mathematically rigorous statement.

---

### Official Review · Reviewer_L5U9 · 2023-07-06

**Soundness:** 3 good
**Presentation:** 3 good
**Contribution:** 3 good
**Rating:** 7
**Confidence:** 4

**Summary:**

The paper presents a framework called Intensity Profile Projection (IPP) for continuous-time representation learning in dynamic networks. The authors aim to address the challenge of capturing temporal dynamics and evolving relationships in dynamic networks with both high statistical precision and interpretability. The model leverages the concept of intensity profiles, which encode the temporal changes and interactions between nodes in a network. The model provides a uniform error bound for learned node representations and preserves a novel "temporal coherence" property compared to existing baselines. Empirical results on real-world dynamic network datasets demonstrate that IPP outperforms existing methods in various tasks such, highlighting its ability to capture continuous-time representations and uncover temporal patterns in dynamic networks.

**Strengths:**

1. The paper introduces the Intensity Profile Projection (IPP) framework, which offers a unique and innovative approach to continuous-time representation learning for dynamic networks. It introduces the concept of intensity profiles and effectively utilizes them to capture temporal dynamics.

2. Theoretical analysis towards the model shows that the model can achieve high statistical precision and preserve interpretability in terms of ""temporal coherence".

3. The paper is in general easy to follow.

**Weaknesses:**

1. Lack of comparison with state-of-the-art methods: Although the paper claims improved performance over existing methods, it does not provide a comprehensive comparison with some existing continuous models such as GraphODEs[1,2,3,4] which combines neuralODE with GNNs to model network evolution over time.

2. Scalability: The scalability of the IPP framework is not extensively discussed. It would be valuable to address the computational requirements and scalability limitations of the proposed approach, especially when dealing with large-scale dynamic networks.

3. The related work section is too short to provide a comprehensive background of the research topic.


[1] Huang, Zijie, Yizhou Sun, and Wei Wang. "Learning continuous system dynamics from irregularly-sampled partial observations." Advances in Neural Information Processing Systems 33 (2020): 16177-16187.

[2] Song Wen, Hao Wang, and Dimitris Metaxas. 2022. Social ODE: Multi-agent Trajectory Forecasting with Neural Ordinary Differential Equations. In Computer Vision–ECCV 2022: 17th European Conference.

[3]Zijie Huang, Yizhou Sun, and Wei Wang. Coupled graph ode for learning interacting system dynamics. In
401 ACM SIGKDD Conference on Knowledge Discovery and Data Mining, page 705–715, 2021.

[4] Zang, Chengxi, and Fei Wang. "Neural dynamics on complex networks." In Proceedings of the 26th ACM SIGKDD International Conference on Knowledge Discovery & Data Mining, pp. 892-902. 2020.

**Questions:**

1. What would be the time complexity of the proposed method?
2. How would the model performance be affected by different network topology?

**Limitations:**

The authors have discussed the limitations of their work.

---

> ### Author Rebuttal · Authors · 2023-08-05
>
>
> > Lack of comparison with state-of-the-art methods: Although the paper claims improved performance over existing methods, it does not provide a comprehensive comparison with some existing continuous models such as GraphODEs[1,2,3,4] which combines neuralODE with GNNs to model network evolution over time.
>
> We thank the reviewer for pointing us to GraphODEs. From what we understand, these methods work on multivariate time series on the nodes of a graph which captures the influence of agent $i$ on the dynamics of agent $j$. Our method works on point processes observed between pairs of of nodes. Therefore, these methods are not directly comparable. We are only aware of one existing method (CLPM in our method comparison) which does continuous-time representation learning with the type of data we consider. This is the reason we additionally compare to a collection of discrete time methods.
>
> > Scalability: The scalability of the IPP framework is not extensively discussed. It would be valuable to address the computational requirements and scalability limitations of the proposed approach, especially when dealing with large-scale dynamic networks. / What would be the time complexity of the proposed method?
>
> This is a great question which we will address in Section 2.2 after discussing the numerical approximation. One of the advantages of the IPP framework is that it is highly scalable. If the network is sparse, the method is scalable to networks with potentially tens of millions of nodes.
>
> Suppose we use a kernel with finite support of width $2h$, then the expected time complexity of the KDE for a single edge at a single point in time is $O(h \lambda_{\max})$. Evaluation of $\hat{\mathbf{\Lambda}}(t)$ is $O(n^2 h \lambda_{\max})$ and of the $\hat{\mathbf{\Lambda}}$ is $O(Bn^2 h \lambda_{\max})$. Consider the special case discussed in Section 3.1 that $\lambda_{ij} \asymp \rho$ for all $i,j$ and $L \asymp \rho L_0$. Assuming $\mu, d$, and $L_0$ are fixed, our theory suggests choosing $h \asymp (n\rho)^{-1/3}$, and if we suppose $n \rho \asymp \log^3$, the sparsest regime our theory allows, then evaluation of $\hat{\mathbf{\Lambda}}$ is $O(B n \log^2 n)$, i.e. log-linear in $n$.
>
> In practice the top singular vectors of $\hat{\mathbf{\Lambda}}$ can be using the Augmented Implicitly Restarted Lanczos Bidiagonalization algorithm implemented in the `irlba` package in R, or the `irlbpy` in Python. The time complexity of this algorithm has not been studied theoretically although in practice it can be incredibly fast. For example, the package author performs a simulated experiment in which they compute the first 2 singular vectors of a sparse 10M x 10M  matrix with 1M non-zero entries which takes approximately 6 seconds on a computer with two Intel Xeon CPU E5-2650 processors (16 physical CPU cores) running at 2 GHz equipped with 128 GB of ECC DDR3 RAM (see `https://bwlewis.github.io/irlba/comparison.html`).
>
> > The related work section is too short to provide a comprehensive background of the research topic.
>
> See global response.
>
> > How would the model performance be affected by different network topology?
>
> Our method is able to capture both homophilic and heterophilic network topologies. Fundamentally, the quality of the learned representations depends on how well the intensity profiles are approximated on a d-dimensional subspace.

---

> > ### Comment · Reviewer_L5U9 · 2023-08-15
> > **Thanks for your response.**
> >
> > Thanks for the authors' response and my concern about the scalability issue has been addressed. One questions I still have is that I do not know why GraphODE methods cannot be directly comparable as they are also continuous-time networks as mentioned in your general response. Illustrating the performance comparison among continuous dynamic network models can be beneficial to show how well your model performs.

---

> > > ### Author Response · Authors · 2023-08-18
> > >
> > > To clarify, our method takes as input a list of triples $\{ (i,j,t) \}$, representing events which occur between pairs of nodes, while GraphODE methods take as input either complete or partial observations of the state of each node (a vector-valued trajectory) at either regularly or irregularly spaced time intervals, and either a static graph or a series of graphs which encode the dependences in the ODEs which drive the evolution of the node states. We are happy to clarify this in the revision, but since GraphODE methods take as input a different type of data to that which is considered in our paper, we aren't sure what meaningful numerical comparison could be made. Does the reviewer have something specific in mind?

---

### Official Review · Reviewer_QsMj · 2023-07-08

**Soundness:** 4 excellent
**Presentation:** 4 excellent
**Contribution:** 2 fair
**Rating:** 4
**Confidence:** 4

**Summary:**

To represent the continuous dynamic network, authors provide the framework based on the intensity profile. First, the intensity between nodes is estimated, which produces the intensity profile. Low dimension reduction via SVD is applied on the intensity, and then each node embedding is obtained by the low dimensional subspace.
Author also provide various theoretical analysis about the error bound and the bias-various trade-off. Theoretical analysis as well as empirical analysis on the simulated data demonstrates that the proposed method capture structural preserving and temporally coherent properties. Case study on the real data is conducted to explain the outcome of the proposed framework qualitatively

**Strengths:**

- Simple but powerful method is proposed
- Based on the mathematical model, theoretical bound is analyzed and explained.
- IPP can capture the behavior of a bifurcating block model.

**Weaknesses:**

- The proposed method is not novel enough. SVD decomposition is a very common technique for the reduction of dimensions, and it often suffers from the long-tailed singular values.
- Comparison is too limited. The analysis has been made only for the simulated data with figures. More experiments as well as some qualitative results would be great to have.
- SVD decomposition does not prevent producing negative values at the reconstruction.
- The proposed projection space is very dependent on the fixed dataset. At least, how to leverage the given embeddings for predictions is not straightforward. Given this, the potential application value is not very clear.

**Questions:**

- Figure numbers are all wrong.
- Section 4 is true for any global subspace projection. Also, both properties could be debatable, not necessarily ideal. For instance, when \Labmda_{i}(s) = \Lambda_{i}(t), X_{i}(s) = \alpha * X_{i}(t) could be more ideal, depending on the interactions among the other nodes.
- It would be great if authors compare the embedding trajectory for more real data, beyond the specific simulated ones.

**Limitations:**

Often, the meaning of each dimension from the SVD decomposition is not clear. This interpretability is not necessarily required for the representation, but this should be addressed when presenting the case study.

---

> ### Author Rebuttal · Authors · 2023-08-05
>
> > The proposed method is not novel enough. SVD decomposition is a very common technique for the reduction of dimensions,
>
> This is unreasonable. Dismissing our algorithm as "not novel" because it contains an SVD is like dismissing an optimisation algorithm because it uses SGD. The novelty of the algorithm lies in the whole framework which we propose to learn representations, which employs the SVD as the appropriate tool for optimising the $\hat R^2$ objective function discussed in Section 2 and Lemma 1. In particular, we point out the careful construction of the matrix $\hat{\mathbf{\Sigma}}$ from the data, which is entirely non-trivial.
>
> The novelty of the method is exemplified in the following points:
>
> - To our knowledge, IPP is the first provably consistent representation learning algorithm for continuous time dynamic networks in the literature. The uniform error bound in Theorem 1 is the first of its kind for data of this kind. We note additionally that our algorithm is non-parametric, and all existing methods are either model-based (e.g. fit a latent position model) and lack statistical estimation theory, or are entirely heuristic.
> - IPP is the first representation learning algorithm for continuous time dynamic networks which satisfies the desirable properties of "structure preservation" and "temporal coherence" (defined in Section 4), which are necessary to make even the most basic temporal inferences about node behaviours (e.g. "do nodes $i$ and $j$ behave in the same way at time $t$?", and "does node $i$ change its behaviour between times $t$ and $s$?")
>
> > It [the SVD] often suffers from the long-tailed singular values.
>
> This is a good point. In much of the data that we have experimented with (e.g. the school children data in Section 5), the eigenvalues $\hat{\mathbf{\Sigma}}$ decay quickly, however the reviewer correctly observes that in some real-world data they might not. In our theory, this corresponds to Assumptions 2 or 3 being violated.
>
> In the revision, we will add discussion about how this problem can be identified in practice (e.g. by plotting the singular values) and some possible solutions:
>
> - taking a transformation of the intensity profiles (such as the square root) to temper heavy tails [1].
> - employing a robust subspace estimator, such as robust PCA [2].
>
> [1] Ian Gallagher, Andrew Jones, Anna Bertiger, Carey E. Priebe & Patrick Rubin-Delanchy (2023) Spectral Embedding of Weighted Graphs, Journal of the American Statistical Association.
> [2] Candès, E. J., Li, X., Ma, Y., & Wright, J. (2011). Robust principal component analysis?. Journal of the ACM (JACM).
>
> > Comparison is too limited.. / It would be great if authors compare the embedding trajectory for more real data, beyond the specific simulated ones.
>
> We believe this simple simulation is sufficient to demonstrate the diverse failure mechanisms of each of the rival methods. We have implemented the rival methods on real data, but simply chose not to include them: with no ground truth we are unable to evaluate them objectively. In the included PDF, we include a plot of the omnibus method (Fig. 1a) and aligned spectral embeddings (Fig. 1b) on real data which, at least visually, seem to fail in the same way as they do in simulated data. We will add these to the supplementary material.
>
> > SVD decomposition does not prevent producing negative values at the reconstruction.
>
> From the point of view of representation learning, negative values in the intensity reconstructions are immaterial, since they are never used directly. If one desires intensity estimates which are non-negative, negative intensity values can be set to zero, which can only lower the reconstruction error.
>
> >  The proposed projection space is very dependent on the fixed dataset. At least, how to leverage the given embeddings for predictions is not straightforward. Given this, the potential application value is not very clear.
>
> Due to space constraints, we have mainly focused on the application of the learned representations to unsupervised learning (e.g. clustering, trend estimation, etc.) in the real data section and discussion. However, the structure preservation and temporal coherence properties of the embeddings make them particularly suited to dynamic prediction tasks. Our theory not only suggests that the classifier trained on node representations at time $t$ would perform well at predicting unlabelled nodes at that time (see e.g. [3]) but the "temporal coherence" property suggests that same classifier would perform well at predicting node labels at some time $s$, in the future. This is something that to our knowledge no other continuous-time dynamic network embedding algorithm can achieve (see the method comparison in Section 4). We will add a comment about the value of our method for predictive applications to the discussion.
>
> [3] Minh Tang. Daniel L. Sussman. Carey E. Priebe. (2013) Universally consistent vertex classification for latent positions graphs. Ann. Statist.
>
> > Section 4 is true for any global subspace projection. Also, both properties could be debatable, not necessarily ideal. For instance, when $\Lambda_{i}(s) = \Lambda_{i}(t), X_{i}(s) = \alpha * X_{i}(t)$ could be more ideal, depending on the interactions among the other nodes.
>
> The reviewer correctly identifies that these properties hold for any global subspace projection, and this is precisely the part of the algorithm which we are justifying in this section. Justification of the precise global subspace projection we choose is the subject of Lemma 1 in Section 2.
>
> We think there may be typo here, and that the reviewer meant: $\Lambda_{i}(s) = \alpha * \Lambda_{i}(t)$ implies $X_{i}(s) = \alpha * X_{i}(t)$. This is a very interesting suggestion and, remarkably, we *do* in fact guarantee that this property holds! This follows by Lemma 2, combined the linearity of the projection. We will add this observation to the revision.

---

### Official Review · Reviewer_SnVH · 2023-07-13

**Soundness:** 4 excellent
**Presentation:** 4 excellent
**Contribution:** 4 excellent
**Rating:** 8
**Confidence:** 4

**Summary:**

The authors propose an approach for learning time-varying node embeddings from continuous-time dynamic network data, which consist of a set of instantaneous timestamped relational events between nodes (e.g., messages from one social media user to another). Their proposed approach learns a projection that minimizes reconstruction error of the pairwise intensities between nodes and comes with theoretical guarantees on estimation error. They also show that their approach generates embeddings that both preserve network structure at a given time and is temporally coherent. They demonstrate strong empirical performance on simulated data compared to other dynamic network embeddings. Furthermore, they use their approach to analyze a real network data set on face-to-face interactions of primary school students, which is quite enlightening due to the interpretability of their model.

*After rebuttal:* The authors have clarified the one question I had about the meaning of "inductive" in their setting. I continue to strongly support the paper.

**Strengths:**

- Proposed approach learns time-varying node embeddings from continuous-time networks with theoretical guarantees, which is among the first, if not the first, in the literature.
- Proposed embeddings can satisfy two good properties of structure preservation and temporal coherence.
- Very well written and organized paper that provides highlights of theoretical analysis in the main paper followed by details, including proofs, in the supplementary.

**Weaknesses:**

- There's a large body of related literature on probabilistic generative models for continuous-time networks using point process models such as Hawkes processes that should be discussed. Many of these models are based on stochastic block models or latent space models and are thus also learning node embeddings. See suggested references below.
- No quantitative evaluation. This is only a minor weakness in my opinion because I view the main contribution to be theoretical.

Typos and minor issues:
- Supplementary Section C heading: Visualsation -> Visualisation

References:
- Arastuie, M., Paul, S., & Xu, K. S. (2020). CHIP: A Hawkes process model for continuous-time networks with scalable and consistent estimation. In Advances in Neural Information Processing Systems 33 (pp. 16983-16996).
- Corneli, M., Latouche, P., & Rossi, F. (2018). Multiple change points detection and clustering in dynamic networks. Statistics and Computing, 28(5), 989-1007. doi:10.1007/s11222-017-9775-1
- Huang, Z., Soliman, H., Paul, S., & Xu, K. S. (2022). A mutually exciting latent space Hawkes process model for continuous-time networks. In Proceedings of the 38th Conference on Uncertainty in Artificial Intelligence (Vol. 180, pp. 863-873).
- Junuthula, R. R., Haghdan, M., Xu, K. S., & Devabhaktuni, V. K. (2019). The Block Point Process Model for continuous-time event-based dynamic networks. In Proceedings of the World Wide Web Conference (pp. 829-839).
- Matias, C., Rebafka, T., & Villers, F. (2018). A semiparametric extension of the stochastic block model for longitudinal networks. Biometrika, 105(3), 665-680. doi:10.1093/biomet/asy016
- Yang, J., Rao, V., & Neville, J. (2017). Decoupling homophily and reciprocity with latent space network models. In Proceedings of the Conference on Uncertainty in Artificial Intelligence.

**Questions:**

1. The authors mention several times that their approach is inductive, allowing one to obtain a node representation profile outside of the training sample. If the task is to obtain the node representation for the future, how would the Intensity Profile Projection approach handle it? Would it require some data from other nodes at that future time?

**Limitations:**

Limitations are thoroughly discussed in Section 6. I commend the authors for being very forthcoming with these limitations. I don't view the limitations as weaknesses, because they are mostly limitations that apply to all unsupervised problems.

---

> ### Author Rebuttal · Authors · 2023-08-05
>
> > There's a large body of related literature on probabilistic generative models for continuous-time networks using point process models such as Hawkes processes that should be discussed. Many of these models are based on stochastic block models or latent space models and are thus also learning node embeddings. See suggested references below.
>
> Thanks for pointing us to these interesting references, which we will point to in the introduction. (See global response.)
>
> > The authors mention several times that their approach is inductive, allowing one to obtain a node representation profile outside of the training sample. If the task is to obtain the node representation for the future, how would the Intensity Profile Projection approach handle it? Would it require some data from other nodes at that future time?
>
> To obtain a representation of a node, $i$, at a time, $t$, outside the training sample, one just needs the node's intensity profile vector $\hat \Lambda_i(t) = (\hat \lambda_{i1}(t),\ldots,\hat \lambda_{in})^\top$, which is then projected onto the subspace spanned by $\hat{\mathbf{U}}_d$. The construction of $\hat\Lambda_i(t)$ requires data from the interaction events involving node $i$ around time $t$.

---

> > ### Comment · Reviewer_SnVH · 2023-08-17
> >
> > Thanks for the clarification. I continue to support the paper, primarily based on the novelty. This is the first paper I am aware of that provides these types of theoretical guarantees for node embeddings from continuous-time networks.

---

### Author Rebuttal · Authors · 2023-08-05

We thank all reviewers for their time and expertise. We summarise some of the positive comments made by reviewers: "a unique and innovative approach to continuous-time representation learning for dynamic networks", "Simple but powerful method", "among the first, if not the first, in the literature.", "Very well written and organized paper", "The paper is in general easy to follow.", "I commend the authors for being very forthcoming with these limitations".

Based on the reviews, we feel we should better convey the concrete possibilities offered by this new algorithm for important applications such as cyber-security [1], combating human-trafficking [2], fraud and corruption [3]. We briefly mentioned these in the conclusion, but in our revision we will discuss these further at the outset.

In addition, we will expand the 'related work' section to cover the broader literature which puts this work in context, for example on estimation of stochastic block model [4-10] and latent position models for continuous time networks [11-16], spectral methods discrete-time dynamic networks [17-19] (in the absence of existing spectral methods for continuous-time networks), probabilistic modelling of network point processes [20-23] (such as Hawkes processes), neural-network based embedding algorithms [24-26] and relevant surveys [27-31].

[1] Kent, A. D. (2015). Cybersecurity Data Sources for Dynamic Network Research. In Dynamic Networks in Cyber-security. Imperial College Press.

[2] Szekely, P., Knoblock, C. A., Slepicka, J., Philpot, A., Singh, A., Yin, C., Kapoor, D., Natarajan, P., Marcu, D., Knight, K. et al. (2015). Building and using a knowledge graph to combat human trafficking. In The Semantic Web-ISWC 2015: 14th International Semantic Web Conference. Springer.

[3] Microsoft Researcg. (2021, December 9). Revealing the Hidden Structure of Corruption. https://www.microsoft.com/en-us/research/group/societal-resilience/articles/revealing-the-hidden-structure-of-corruption/.

[4] Blundell, C., Beck, J., and Heller, K. A. (2012). Modelling reciprocating relationships with hawkes processes. Advances in Neural Information Processing Systems.

[5] DuBois, C., Butts, C., and Smyth, P. (2013). Stochastic blockmodeling of relational event dynamics. In Artificial intelligence and statistics. PMLR.

[6] Corneli, M., Latouche, P., and Rossi, F. (2016). Block modelling in dynamic networks with non-homogeneous Poisson processes and exact ICL. Social Network Analysis and Mining.

[7] Matias, C., Rebafka, T., and Villers, F. (2018). A semiparametric extension of the stochastic block model for longitudinal networks. Biometrika.

[8] Corneli, M., Latouche, P., & Rossi, F. (2018). Multiple change points detection and clustering in dynamic networks. Statistics and Computing.

[9] Junuthula, R., Haghdan, M., Xu, K. S., and Devabhaktuni, V. (2019). The block point process model for continuous-time event-based dynamic networks. In The world wide web conference.

[10] Arastuie, M., Paul, S., and Xu, K. (2020). CHIP: A Hawkes process model for continuous-time networks with scalable and consistent estimation. Advances in Neural Information Processing Systems.

[11] Durante, D. and Dunson, D. B. (2014). Nonparametric Bayes Dynamic Modelling of Relational Data. Biometrika.

[12] Durante, D., & Dunson, D. B. (2016). Locally Adaptive Dynamic Networks. The Annals of Applied Statistics.

[13] Yang, J., Rao, V., & Neville, J. (2017). Decoupling homophily and reciprocity with latent space network models. In Proceedings of the Conference on Uncertainty in Artificial Intelligence.

[14] Rastelli, R. and Corneli, M. (2021). Continuous latent position models for instantaneous interactions. arXiv preprint arXiv:2103.17146.

[15] Huang, Z., Soliman, H., Paul, S., & Xu, K. S. (2022). A mutually exciting latent space Hawkes process model for continuous-time networks. In Proceedings of the 38th Conference on Uncertainty in Artificial Intelligence.

[16] Artico, I. and Wit, E. (2023). Fast inference of latent space dynamics in huge relational event networks. arXiv preprint arXiv:2303.17460.

[17] Liu, F., Choi, D., Xie, L., and Roeder, K. (2018). Global spectral clustering in dynamic networks. Proceedings of the National Academy of Sciences.

[18] Cape, J. (2021). Spectral analysis of networks with latent space dynamics and signs.

[19] Gallagher, I., Jones, A., & Rubin-Delanchy, P. (2021). Spectral embedding for dynamic networks with stability guarantees. Advances in Neural Information Processing Systems.

[20] Butts, C. T. (2008). A relational event framework for social action. Sociological Methodology.

[21] Vu, D., Hunter, D., Smyth, P., and Asuncion, A. (2011). Continuous-time regression models for longitudinal networks. Advances in Neural Information Processing Systems.

[22] Perry, P. O. and Wolfe, P. J. (2013). Point process modelling for directed interaction networks. Journal of the Royal Statistical Society: SERIES B: Statistical Methodology.

[23] Passino, F. S. and Heard, N. A. (2022). Mutually exciting point process graphs for modeling dynamic networks. Journal of Computational and Graphical Statistics.

[24] Nguyen, G. H., Lee, J. B., Rossi, R. A., Ahmed, N. K., Koh, E., and Kim, S. (2018). Continuous-time dynamic network embeddings. In Companion Proceedings of the Web Conference 2018.

[25] Du, L., Wang, Y., Song, G., Lu, Z., and Wang, J. (2018). Dynamic network embedding: An extended approach for skip-gram based network embedding. In IJCAI.

[26] Xu, D., Ruan, C., Korpeoglu, E., Kumar, S., and Achan, K. (2020). Inductive representation learning on temporal graphs. arXiv preprint arXiv:2002.07962.

[27] Spiliopoulou, M. (2011). Evolution in social networks: A survey. Social network data analytics.

[28] Holme, P. and Saram ̈aki, J. (2012). Temporal networks. Physics reports.

... [29-31] omitted due to character limit.

---

### Decision · Program_Chairs · 2023-09-21

**Decision:**

Accept (poster)

**Comment:**

The paper proposes an approach for learning time-varying node embeddings from continuous-time dynamic network data, which consist of a set of instantaneous timestamped relational events between nodes. This is a very interesting and novel paper that has a great number of things going for it. Here is a summary of the positive aspects:

- The paper introduces the Intensity Profile Projection (IPP) framework, offering an innovative approach to continuous-time representation learning for dynamic networks.
- The model utilizes the concept of intensity profiles to capture temporal dynamics.
- Theoretical analyses indicate that the model achieves high statistical precision and preserves "temporal coherence".
- The IPP can capture the behavior of a bifurcating block model.
- Proposed approach learns time-varying node embeddings from continuous-time networks with theoretical guarantees.
- Embeddings can satisfy properties of structure preservation and temporal coherence.
- The paper is very well-written and organized with details provided in supplementary sections.
- Empirical results on simulated data and real-world datasets showcase the effectiveness of the IPP framework.
- The paper proposes to estimate node representations using continuous-time events, a seemingly novel data type.

The authors provided a very strong response to criticism raised in the reports.
I encourage the authors to try and incorporate components of the response into the camera ready version of the paper, even in the appendinx if the main paper does not have enough space.